# Evolving Prompts In-Context: An Open-ended, Self-replicating Perspective

**Jianyu Wang** [1 2]   **Zhiqiang Hu** [1 2]   **Lidong Bing** [3]

## Abstract

We propose a novel prompt design paradigm that challenges conventional wisdom in large language model (LLM) prompting. While conventional wisdom prioritizes well-crafted instructions and demonstrations for in-context learning (ICL), we show that pruning random demonstrations into seemingly incoherent "gibberish" can remarkably improve performance across diverse tasks. Notably, the "gibberish" always matches or surpasses state-of-the-art automatic prompt optimization techniques, achieving substantial gains regardless of LLM alignment. Nevertheless, discovering an effective pruning strategy is non-trivial, as existing attribution methods and prompt compression algorithms fail to deliver robust results, let alone human intuition. In terms of this, we propose a *self-discover* prompt optimization framework, PROMPTQUINE, an evolutionary search framework that automatically searches for the pruning strategy by itself using only low-data regimes. Much like the emergent complexity in nature—such as symbiosis and self-organization—arising in response to resource constraints, our framework evolves and refines unconventional yet highly effective prompts by leveraging only the tokens present within the context. We demonstrate its effectiveness across classification, multi-choice question answering, generation and math reasoning tasks across LLMs, while achieving decent runtime efficiency. We hope our findings can guide mechanistic studies on in-context learning, and provide a call to action, to pave the way for more open-ended search algorithms for more effective LLM prompting.

This work was partially done when Lidong Bing was at DAMO Academy, Alibaba Group. [1]DAMO Academy, Alibaba Group [2]Hupan Lab [3]MiroMind. Correspondence to: Jianyu Wang <jiw102@ucsd.edu>.

*Proceedings of the 42nd International Conference on Machine Learning*, Vancouver, Canada. PMLR 267, 2025. Copyright 2025 by the author(s).

## 1. Introduction

Prompting large language models (LLMs) has become the *de facto* standard for numerous applications, shifting the community focus to designing prompts that maximize model performance. However, this task is inherently complex due to the nuanced and often unpredictable behavior of LLMs (Lu et al., 2022; Liu et al., 2023). Subtle changes in phrasing, structure, or context can dramatically affect outputs (Jiang et al., 2020; Shi et al., 2023). Consequently, prompt engineering relies heavily on iterative experimentation and evaluation. On the other hand, automatic prompt optimization (Liu et al., 2023) explores to minimize the human involvement by leveraging computations to refine prompts iteratively. The conventional wisdom (Zhao et al., 2021; Lu et al., 2022; OpenAI, 2023; Wan et al., 2024) suggests that well-specified task instruction, combined with a few tuned demonstrations for ICL, yields the best results.

This paper presents a study that challenges conventional wisdom by showing that pruning clear, detailed demonstrations into seemingly incoherent "gibberish" (both syntactically and semantically strange) can, counterintuitively, improve performance across various tasks. Notably, this effect generalizes across models, regardless of alignment (Shen et al., 2023b), suggesting a broader misspecification of unnatural language in current LLMs. Even more surprisingly, we find that this "gibberish" consistently matches or surpasses the performance of state-of-the-art automatic prompt optimization results in several tasks. Consequently, we propose a novel conceptual framework that reframes prompt compression as guided prompt search, enhancing both serving efficiency and task performance. We further explore algorithms to achieve these improvements.

To derive effective pruning strategies, one might expect existing instance attribution (Li et al., 2016a; Yin & Neubig, 2022) or prompt compression methods (Li et al., 2023; Jiang et al., 2023c;d; Pan et al., 2024) to provide guidance. However, we find that none of these methods can reliably produce accurate token importance scores to guide the pruning, let alone human intuition according to word semantics. A more practical idea is to ask algorithms to discover the pruning strategies by themselves. We thus develop a *self-discover* prompt optimization framework, Genetic Prompt-Quine (PROMPTQUINE), an evolutionary search framework

that automatically searches for the pruning strategy by itself using low-shot regimes. The philosophy of this framework is essentially a *self-replicating* program (Von Neumann et al., 1966) that copies and mutates the prompt itself (e.g., pruning random tokens). The mutated prompts compete for limited resources to survive based on their fitness, with only the most performant surviving, thereby evolving effective pruning strategies over multiple generations. The entire process closely resembles the evolutionary dynamics of biological systems, where self-replication and adaptation drive the emergence of more effective strategies over successive generations (i.e., evolutionary self-replication).

Evolutionary self-replication, a fundamental concept in Darwinian evolution (Ofria & Wilke, 2004), explains how life reproduces and evolves through genetic variation and natural selection, fostering the emergence of unpredictable traits and behaviors that enhance adaptability to constantly shifting environmental conditions. Similarly, the prompt design problem can also be framed as an evolutionary process, where prompts shall be iteratively refined to adapt to complex LLM environments. As such, prompts that are optimal for complex LLMs may exceed human intuition and require methods beyond manual design. Inspired by our findings, we propose embracing more open-ended innovations (or broader open-endedness (Stanley et al., 2017))—shifting from the human language space towards the "LLM language space"—to advance LLM prompting strategies.

We demonstrate the effectiveness of our search framework through large-scale experiments across tasks and LLMs. Our results show that **pruning a low-shot ICL prompt could perform comparably to state-of-the-art methods on a range of tasks**, while maintaining competitive runtime efficiency compared to prior optimization methods (e.g., Table 10 in Appendix). Notably, our PROMPTQUINE framework is inherently well-suited for parallelization, which can further enhance both its scalability and efficiency, for example by parallelizing reproduction and fitness evaluation. Moreover, we find that the task improvements of our approach could becosme more pronounced as the number of in-context examples increases, suggesting that scaling to more shots can unlock additional performance gains. This suggests that richer prompt variations can drive further gains. Finally, we show that key findings on label word importance also hold for our ICL pruning, with the additional insight that **pruning has potentials to enhance performance with random verbalizers, even when starting from chance**.

## 2. Problem Formalisms

### 2.1. Preliminaries: In-context Learning

ICL describes an emergent capability of LLMs that given a few training examples appended in context, the LLM is able

to be conditioned to infer the task results. Formally, given K input-label pairs $\{(\mathbf{x}_i, \mathbf{y}_i)\}_{i=1}^{K}$ and the task test input $\mathbf{x}_{\text{test}}$ concatenated in the input context, the LLM is conditioned to generate the task prediction:

$$\mathbf{y}_{\text{test}} \sim \mathcal{P}_{LM}\left(\cdot \mid \mathbf{x}_1, \mathbf{y}_1, \ldots, \mathbf{x}_K, \mathbf{y}_K, \mathbf{x}_{\text{test}}\right) \quad (1)$$

users can then parse the task output from the prediction $\mathbf{y}_{\text{test}}$ (i.e., mapping into corresponding task verbalizers (Schick & Schütze, 2021)).

### 2.2. The Motivating Discussion

An intriguing observation in recent LLM prompting research (Shin et al., 2020; Deng et al., 2022; Zou et al., 2023) suggests that, for certain tasks, unintelligible or unnatural prompts (e.g., `StaffAreaFocusHardware Advisory` for News Classification) can outperform carefully crafted natural language instructions. This unnatural language phenomenon has been discussed as a form of *secret language* (Daras & Dimakis, 2022) in literature.

Specifically, *secret language* often refers to unnatural language prompts whose syntax and semantics are incoherent and difficult for humans to parse, yet can be surprisingly effective in certain scenarios. In the absence of theoretical explanation for their emergence, and universally effective method for large-scale discovery, such prompts, either discovered by coincidence or found after extensive computation by certain algorithms (Shin et al., 2020; Deng et al., 2022; Jones et al., 2023, *inter alia*), such as the news classification prompt we discussed above, are typically regarded as mysterious, hidden, and inherently non-scalable.

We argue that such seemingly chaotic discoveries may actually contain universal insights into LLM sensitivity towards prompt design. We detail our thought process as follows: It's counter-intuitive that unnatural prompts can outperform natural instructions, despite being extensive trained to align human language. This suggests LLMs may only experience *superficial alignment* (Greenblatt et al., 2024) and instead may prioritize hypotheses over the explicit structure of human linguistics. Recently, Chan et al. (2022) identified that transformer language models, especially LLMs, exhibit sparse, rule-based generalization in ICL, where minimal features can dominate predictions (Dasgupta et al., 2022). This raises the possibility that some input features can be redundant or inessential towards task prediction. We are thus curious whether natural language prompts, e.g., ICL, could be improved simply by removing certain input features in-context, exploring the potential of ICL optimization in-context. In other words, we deliberately explore disrupting the grammatical structure of prompts in an attempt to approach a structure that LLMs might prefer.

Alternatively, such perturbation might be viewed as a nearby

search within semantically coherent natural language contexts, potentially outperforming the unnatural artifacts from token-level searches with limited algorithmic capacity (Shin et al., 2020; Deng et al., 2022). We refer to this as the *Partial Context Hypothesis*.

Specifically, given a natural language prompt, e.g., ICL prompt, $\mathbf{x} = (x_1, x_2, \ldots, x_n)$ with task performance $\mathbf{X}$, is it possible to prune a few prompt tokens, resulting in a pruned prompt $\mathbf{z} = (z_1, z_2, \ldots, z_m), m \leq n$, with significantly improved task performance $\mathbf{Z}$. Notably, the prompt can reach or even surpass task performance $\mathbf{Y}$ of unnatural language prompts $\mathbf{y}$ discovered by prior token-level search algorithms (Shin et al., 2020; Deng et al., 2022), although initial performance $\mathbf{X}$ may be substantially lower.

As we demonstrate in Appendix B through a pilot study using a simple hill-climbing approach (Section 3.1), the hypothesis proves effective and may outperform the unnatural language prompts discovered by Deng et al. (2022) across various contexts—especially in pruning ICL, which is our focus. In terms of the effectiveness, for example, the popular prompt "Let's work this out step by step to be sure we have the right answer," introduced by Zhou et al. (2022), improves InstructGPT's performance on the MultiArith dataset (Roy & Roth, 2015) from 78.7% to 81.5%, outperforming the earlier prompt "Let's think step by step" (Kojima et al., 2022). By pruning the prompt to "Let's work out step by step sure we right answer", we achieve an even higher result of 86.7% (Appendix, Table 7).

### 2.3. Compression as Guided Search: A Reformulation

The *prompt compression* is a widely studied field where the typical target is to improve the inference speed (Li et al., 2023; Jiang et al., 2023d;b; Pan et al., 2024). In contrast, we frame the prompt compression problem as a *guided prompt search* where the task is construed as searching for the prompt subsequence which can elicit improved task results. To avoid the ambiguity, we mainly use the term "prompt pruning" in the paper.

Formally, we describe the search problem as below:

Given input prompt $\mathbf{x} = (x_1, x_2, \ldots, x_n)$, the goal is to locate a pruned prompt $\mathbf{z} = (z_1, z_2, \ldots, z_m), m \leq n$ as a subsequence of input prompt $\mathbf{x}$. The subsequence length $m$ is not predefined and it shall continuously adjust over optimization. Our search optimizes a non-differentiable task objective $f(\mathbf{z}; \mathbf{x}, \mathcal{D})$ which typically represents the task performance over the dataset $\mathcal{D}$ and aims at returning the optimal solution, i.e., an optimally pruned prompt we discovered so far. In the remainder of this section, we will describe the search space, the search objective as well as the overall principles for the solution selection in detail.

**Search Space.** The search space refers to all possible solution candidates that the algorithm can explore. In our context, it specifically refers to any prompt subsequences extracted from the original prompts. Ideally, the word order within the prompts could also be altered during optimization. However, this operation would significantly complicate the search problem, resulting in an exponentially larger search space. Therefore, in this paper, we focus primarily on the fixed-order prompt subsequence search.

**Search Objective.** The search objective functions as a performance measure to evaluate the quality of candidate solutions, especially assessing the effectiveness of prompts in enhancing downstream task performance. For instance, prompt quality can be evaluated using an aggregated metric (e.g., classification accuracy) on a held-out set separated from the original dataset. Since the search objective is typically non-differentiable, we cannot just approximate the solutions via traditional gradient ascent.

**Solution Selection.** Once the search converges or terminates, the algorithm returns a selected optimal solution, i.e., an optimal prompt. The optimality of a prompt is typically measured by an aggregated metric on a held-out dataset. This dataset is often referred to as the validation set, with final performance reported separately on a test set using the task-specific metric. We ensure strict separation between the validation and test sets to prevent data leakage and enable a reliable assessment of generalization.

## 3. PROMPTQUINE

We now introduce our search framework PROMPTQUINE. We'll begin with a straightforward hill-climbing search in Section 3.1, which serves as a strong baseline with good empirical performance and is also the method used in our pilot study (Appendix B) to validate the *Partial Context Hypothesis*. Then, we outline the core design principles for further improvements in Section 3.2, followed by an exploration of the objective landscape and the justification for using evolutionary search in Section 3.3. Section 3.4 outlines our evolutionary search framework for PROMPTQUINE.

### 3.1. A Simple Hill-climbing Search

Due to the absence of well-established or highly efficient algorithms for this prompt subsequence search, as well as the failure of attribution methods (Appendix C) and prompt compression algorithms (e.g., Table 1), we start with a greedy local search method. Basically, it works by iteratively pruning tokens, removing those that improve the prompt's performance. This continues until no token can be pruned without harming validation set performance. This follows the general framework of hill-climbing. While this

approach may not yield optimal solutions, it provides a quick and intuitive way to explore the hypothesis and gain initial insights, as detailed in Appendix B.

Specifically, we primarily adopt the Threshold Accepting (TA) algorithm (Dueck & Scheuer, 1990), which builds upon the First-Choice Hill Climbing (FCHC) algorithm (Russell & Norvig, 2016). Namely, given a prompt sequence $\mathbf{x} = (x_1, x_2, \ldots, x_n)$, the algorithm tracks the current solution (initialized as prompt $\mathbf{x}$) and generates a new candidate by making a local per-token change, e.g., removing a token $x_i$. If the new candidate improves upon the current solution, it is accepted, and the tracked solution is updated accordingly. This directly contrasts with Steepest-Ascent Hill Climbing (SAHC) (Russell & Norvig, 2016), which evaluates all possible one-token modifications and selects the one that yields the greatest improvement (called *SAHCPruning*)—our method offers significant speedups by accepting any modification that improves the prompt.

The algorithms continue until no further improvements are found or a stopping criterion is met. To ensure reproducibility, we fix the pruning order (the method used to make per-token changes) in our implementation. In fact, we apply a left-to-right pruning order, iterating over $x_1$ to $x_n$, and proposing a new prompt by removing the most recently visited token, $x_i$, at step $i$. As the algorithm converges when no tokens can be further removed, it follows that we may repeat the left-to-right iterations multiple times, with each iteration initializing the tracked solution using the optimal prompt from the previous iteration. We illustrate the whole idea in Appendix, Algorithm 1. We call this approach *TAPruning*, which helps escape local optima, as observed in preliminary experiments, by further accepting suboptimal candidates within a predefined threshold (e.g., $\geq 96\%$ of the current maximum prompting performance). This suggests a **deceptive nature of our search objective** (i.e., held-out score). We'll revisit this later in Section 3.3 and Section 4, where we explore how greedy hill-climbing struggles, becoming deeply trapped in local optima, and emphasize the need for rewarding *suboptimal stepping stones* (Lehman & Stanley, 2011; Stanley & Lehman, 2015) to global optimization.

As shown in Table 6 (Appendix B) for Meta-Llama-3-8B-Instruct (AI@Meta, 2024), and an extended study on various LLMs across sizes (e.g., Table 10 in Appendix B) using held-out performance for prompt selection, we demonstrate consistent performance gains across models and tasks through pruning. Notably, such improvement is independent of the ICL prompt we sampled, suggesting a potentially effective approach to stabilize ICL performance (Zhao et al., 2021; Lu et al., 2022; Rubin et al., 2022). Besides, while simple, *TAPruning* already delivers competitive results against prior search methods in both task results and runtime efficiency across several tasks (e.g., Table 11 in Appendix D.3),

making it a strong contender for practical use. We explore further improvements in the following sections.

### 3.2. Design Principles for Improved Performance

A search framework consists of three key components: the search space, algorithm, and objective. While refining the search space is constrained by the black-box nature of LLMs, there is significant potential to improve the algorithm and objective: (1) *Search Algorithm*: Hill-climbing may be suboptimal; insights into the search landscape (Deb & Saha, 2010; Ecoffet et al., 2019) can guide better algorithms. (2) *Search Objective*: The high cost of prompt evaluation, driven by task metrics like accuracy, can be reduced by using more expressive proxies (Yang et al., 2024b). We attempt to integrate these principles into our designs.

### 3.3. Navigating Sparse, Multimodal Search Landscapes

We now examine the search challenges in our optimization landscape to inspire more effective algorithms.

To begin with, we revisit our hill-climbing approach (e.g., TA) and highlight a key assumption: hill climbing guarantees convergence to the global optimum only in unimodal spaces (Glover & Kochenberger, 2003). In such space, a single optimum, i.e., a single peak, allows efficient optimization by following increasing objective values without the need for exhaustive searches in multiple directions. This assumption, however, is very likely fail in our context.

As shown in the Figure 1 (*Left*), we relax the left-to-right order constraint in our hill-climbing algorithm (*TAPruning*) and explore random pruning orders in multiple runs using purely greedy hill-climbing. The results indicate that these runs converge to different solutions, leading to significant variations in task performance. This suggests a potentially complex, multimodal nature of the search problem, where traditional hill-climbing—and its variants, such as Simulated Annealing (Bertsimas & Tsitsiklis, 1993) and TA—struggle to perform effectively. These findings highlight the need for more global exploration strategies to address local optima.

To navigate the multimodal landscape, both unstructured methods like Random Search (Zhigljavsky, 2012) (RS) and structured approaches such as Population-based Search (Beheshti & Shamsuddin, 2013) (e.g., evolutionary search, ES) can be used, each offering unique advantages depending on the search space. In certain scenarios, RS can outperform more complex structured methods, particularly when local optima are densely packed (Bergstra & Bengio, 2012), as unstructured approaches may more efficiently explore multiple regions and find better solutions, while population-based approaches may overly exploit a local region.

We investigate the use of RS and population-based search

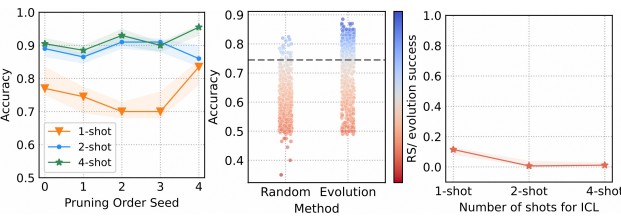

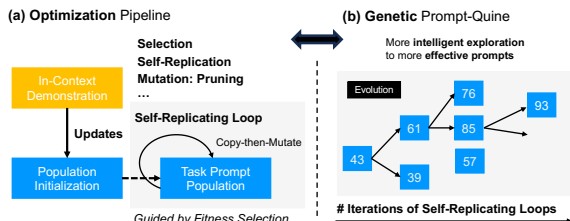

*Figure 1.* Optimization challenges in our ICL-initialized landscape using Llama-3-8B-Instruct for subjectivity classification. **Left**: Randomizing pruning order in hill-climbing search leads to varying task performance, highlighting the multimodal nature. **Middle**: Evolutionary search (ES) outperforms random search (RS) in identifying high-quality solutions, with *TAPruning* result as a dashed line. **Right**: Relative success rate of RS over ES approaches zero as task difficulty increases, particularly when solution sparsity is enhanced by expanding the context. Further studies, with consistent conclusions, are provided in Appendix D.2.

*Figure 2.* Overview of the PROMPTQUINE framework. Similar to natural selection, our framework evolves prompts by copying and mutating them (i.e., pruning random tokens). Guided by task-specific selection pressures, it progressively optimizes itself. Notably, the generation of *unnatural language prompts*, despite introducing unexpected variations, consistently outperforms manually designed prompts, representing a step towards open-ended self-improvement in AI (Schmidhuber, 2003; Nisioti et al., 2024).

(i.e., evolutionary search, ES using our PROMPTQUINE design in Section 4.1) in the ICL landscape, conducting independent sampling runs to generate 1,000 unique prompt samples for both subjectivity classification (Subj) and natural language inference (SNLI). As shown in the Figure 1 (*Middle*), RS proves highly inefficient in obtaining high-quality prompts that outperform *TAPruning*, while ES achieves better performance under constrained prompt samples. Inspired by Real et al. (2020), we define the number of the *acceptable prompts* under the fixed budget as the number of sampled pruned prompts which outperform *TAPruning*'s average performance. The *success rate* is then defined as the number of *acceptable prompts* divided by the number of pruned prompts sampled. As shown in Figure 1 (*right*), the relative success rate of RS to ES (i.e., the *success rate* of RS divided by the *success rate* of ES) approaches zero as task difficulty increases. For example, when dealing with standard long contexts, where high-quality prompts are sparse in the search space, unstructured methods like RS may struggle to efficiently navigate the space. We thus recommend ES, as it is more robust to the search dimensions.

## 3.4. Evolutionary Search for PROMPTQUINE

We now provide an overview of our ES algorithm for prompt subsequence search. Specifically, we use Genetic Algorithm (GA) (Holland, 1992), due to its inherent compatibility with our problem. In this approach, we evolve a population of pruned prompts, where binary token masks serve as *genotypes* and the resulting ICL prompts as *phenotypes*. Mutations (i.e., pruning tokens) are implemented via bit-flip 1-to-0 operations. Elitism-based selection guides offspring survival, enabling the autonomous evolution of pruning strategies. Additional details are provided in Appendix D, along with Algorithm 3. Our GA variants incorporate several designs, which effectively improves the search quality.

We refer the reader to Appendix D.1 for details. Tuning the GA for an unknown landscape requires extensive trial and error. We recommend using our configurations for follow-up experiments and discuss key designs below.

We initialize the entire population with duplicates of ICL prompts, as early experiments with random pruning for initializations showed no significant advantage. For mutation rate, a uniformly random selection of the number of flipped bits among the values $\in \{1, 2, 3, 4\}$ effectively balances the exploration-exploitation. We then use *tournament selection* with slightly reduced selection pressure, sampling k individuals and selecting the best for reproduction, which helps mitigate local optima. Most crucially, we apply *regularized evolution* (Real et al., 2017; 2019), where only new offspring compete for population inclusion. This approach, which we have empirically validated, is highly effective in navigating the ICL landscape, particularly in addressing the premature convergence issue that standard GA struggles to overcome—an issue that is a key bottleneck in tuning the configurations for ICL. As we demonstrate in Appendix D.7, such simple approach outperforms many complex diversity-preserving mechanisms (Friedrich et al., 2009) in balancing search speed and solution quality.

Simple regularized evolution sacrifices some exploration, limiting its effectiveness in broader contexts. To improve exploration, we increase the selection probability for each individual through additional designs, promoting more reproduction in each generation. This approach is guided by two algorithmic frameworks (Syswerda, 1991): a parallelizable Generational GA (GGA) and a more exploratory Steady-state GA (SSGA), detailed in Appendix D.1. Unless otherwise noted, we use SSGA for 1-shot ICL results. Subsequent experiments also reveal that GGA achieves comparable performance across various ICL search landscapes.

Finally, we introduce an additional prompt re-ranking phase,

*Table 1.* Performance of PROMPTQUINE and baseline methods measured by overall accuracy (%) in classification tasks. All methods use Meta-Llama-3-8B-Instruct (AI@Meta, 2024) as the backbone LM for a fair comparison. The best results are highlighted in **bold** and the second best are underlined. Ratio denotes the compression ratio over the original 1-shot ICL prompts.

| Method | SST-2 | Subj | AG's News | Yelp-5 | SNLI | Yahoo | Avg. | Ratio ↑ |
|---|---|---|---|---|---|---|---|---|
| ICL (1-shot, original) (Brown et al., 2020) | 95.9 (0.6) | 66.7 (4.3) | 83.7 (1.9) | 52.2 (6.0) | 61.9 (2.0) | 57.1 (6.9) | 69.6 | 0.0% |
| LLMLingua (Jiang et al., 2023c) | 95.8 (0.6) | 64.0 (4.2) | 85.3 (1.8) | 49.7 (6.6) | 64.2 (1.7) | 49.5 (8.2) | 68.1 | 30.0% |
| LLMLingua2 (Pan et al., 2024) | 61.8 (11.8) | 60.2 (5.1) | 58.1 (13.0) | 45.4 (5.7) | 51.5 (5.1) | 57.1 (4.4) | 55.7 | 71.2% |
| ICL (4-shot) (Brown et al., 2020) | 94.8 (1.5) | 74.0 (7.8) | 86.6 (1.8) | 61.9 (0.6) | 60.0 (2.5) | 54.4 (5.7) | 72.0 | - |
| RLPrompt (Deng et al., 2022) | 88.4 (1.5) | 82.9 (0.5) | 84.7 (0.7) | 48.1 (1.0) | 42.4 (4.2) | 58.5 (0.6) | 67.5 | - |
| EvoPrompt (Guo et al., 2023) | 92.9 (0.2) | 84.1 (0.3) | 86.5 (1.0) | 51.5 (0.6) | 68.2 (0.6) | 58.6 (0.3) | 73.6 | - |
| Promptbreeder (Fernando et al., 2024) | 96.0 (0.4) | 83.6 (3.5) | 88.6 (0.8) | 59.3 (1.6) | 64.2 (1.3) | 62.9 (1.4) | 75.8 | - |
| Promptbreeder (4-shot) (Fernando et al., 2024) | 95.8 (0.5) | 83.1 (3.0) | 88.5 (1.0) | 59.3 (1.5) | 59.6 (1.9) | 65.0 (1.1) | 75.2 | - |
| *TAPruning* (1-shot ICL, Ours) | 95.0 (1.5) | 74.5 (3.9) | 88.6 (0.3) | 60.2 (0.9) | 68.6 (2.9) | 61.7 (1.7) | 74.8 | 60.2 % |
| *SAHCPruning* (1-shot ICL, Ours) | 96.0 (0.7) | 77.3 (6.6) | 88.5 (0.8) | 58.5 (2.2) | 68.4 (2.8) | 62.8 (0.9) | 75.3 | 8.7% |
| **PROMPTQUINE (1-shot ICL, Ours)** | 96.2 (0.2) | 86.5 (2.0) | 89.2 (1.8) | 59.7 (2.1) | 69.2 (2.0) | 64.2 (1.3) | 77.5 | 52.9 % |
| *TAPruning* (4-shot ICL, Ours) | 95.4 (1.5) | 86.9 (0.7) | 88.9 (0.8) | 61.3 (1.6) | 67.3 (1.6) | 63.8 (1.1) | 77.3 | - |
| **PROMPTQUINE (4-shot ICL, Ours)** | **96.4 (0.4)** | **93.1 (0.8)** | **89.4 (1.8)** | **64.3 (0.6)** | **78.6 (3.1)** | **66.2 (1.5)** | **81.3** | - |

called "calibration-then-selection", using the entire held-out score (e.g., what we used for *TAPruning*) to mitigate potential overfitting to the imperfect evaluation proxy. This function refines the prompt rankings, allowing for more accurate identification of the true *elite* prompts. We then select the "optimal" prompt from the calibrated rankings.

## 4. Task Designs & Results

Due to space constraints, we discuss classification and generation in the main paper, and put multi-choice question answering and chain-of-thought reasoning in Appendix D.5 and D.6. In all these tasks, PROMPTQUINE consistently achieves improved results over *TAPruning*.

### 4.1. PROMPTQUINE for Classification

For classification, probability-based prompt selection (Yang et al., 2024b) demonstrates some success in few-shot settings (Lu et al., 2022), enabling fine-grained measurement using metrics like Mutual Information (Sorensen et al., 2022), Entropy (Lu et al., 2022), and Majority Voting (Liao et al., 2022). Through extensive experiments, we finally take the piecewise reward function from (Deng et al., 2022) as our default fitness measure, with details in Appendix D.3. The existence of these multiple metrics can also be extended to multi-objective optimization (Deb et al., 2002) or novelty search (Lehman & Stanley, 2011), leveraging complementary proxies (Vo & Luong, 2024) for prompt selection. However, preliminary experiments indicate that these approaches offer no significant advantages over our single-objective formulation. Consequently, we center our efforts on the single-objective approach in this work.

**Models and Baselines.** We report Meta-Llama-3-8B-Instruct (Llama3B-It) (AI@Meta, 2024) in the main paper,

and leave others in Table 10 (Appendix D.3). We consider the following methods for comparisons: (1) Prompt Compression: LLMLingua (Jiang et al., 2023c) and LLMLingua2 (Pan et al., 2024); (2) Prompt Optimization: RLPrompt (Deng et al., 2022), EvoPrompt (Guo et al., 2023) and Promptbreeder (Fernando et al., 2024). Unless otherwise specified, we append the optimized instructions from EvoPrompt and Promptbreeder to our 1-shot ICL prompts to form their few-shot versions. We report RLPrompt's templates in Appendix B. (3) In-context Learning (ICL): In addition to 1-shot ICL, which directly illustrates the benefits of our pruning methods, we also include 4-shot ICL to assess whether simply increasing the number of shots (Zhao et al., 2021) can easily match the benefits of pruning. Finally, we include *SAHCPruning* on 1-shot ICL as a purely greedy baseline to illustrate the deceptiveness of the landscape. The results are averaged across five seeds, with five different ICL initializations. All optimized prompts are selected based on the same held-out accuracy, with full details in Appendix D.3.

**Evaluation Settings.** We evaluate SST-2 (Socher et al., 2013), Yelp-5 (Asghar, 2016), Subj (Pang & Lee, 2004), AG's News (Zhang et al., 2015), Yahoo (Labrou & Finin, 1999), and SNLI (Bowman et al., 2015) as our Pilot Study (Appendix B). The overall statistics are in Table 5 (Appendix B). As we find 8-shot balanced samples are generally sufficient for fitness estimation, we extract such paired samples from our held-out set (200 validation samples as *TAPruning* in Appendix B) as in-search fitness estimation samples, and use the entire held-out accuracy for prompt re-ranking. We provide additional details and suggestions in Appendix D.3.

**Results.** As shown in Table 1, overall, PROMPTQUINE on 1-shot ICL is able to match or surpass state-of-the-

art performance across settings. One particular baseline we emphasize for comparison is the Promptbreeder (Fernando et al., 2024), which shares a closely aligned spirit with our approach—leveraging evolutionary algorithms to promote open-endedness in the context of LLMs. Their framework, while demonstrating impressive generality, relies more heavily on manual engineering (e.g., handcrafted mutation prompts) and external resources (e.g., new tokens). In contrast, PROMPTQUINE fosters open-endedness under tighter resource constraints, by evolving unnatural language prompts using only tokens within context. We show that PROMPTQUINE also has potentials to be comparable in specific problem contexts. We further compare the runtime efficiency against various baselines in Appendix, Table 11. To the best of our knowledge, this is the first token-level search (*TAPruning* & PROMPTQUINE) capable of optimizing in just minutes. Please refer to Appendix 4.1 for additional analysis, especially for the intriguing observations that *SAHCPruning* may not always outperform *TAPruning*.

## 4.2. PROMPTQUINE for Text Generation

Generation tasks present unique challenges for the limitations of automatic evaluation metrics, such as surface form-based measures (Papineni et al., 2002) and neural embedding-based metrics (Deng et al., 2021). These metrics often fail to capture the full complexity of the task or accurately reflect fine-grained optimization progress, particularly in open-ended generation tasks. In open-ended cases, e.g., topic-based generation, only sparse, qualitative scores are available, e.g., those by LLM-as-a-Judge (Zheng et al., 2023) based on predefined principles, or by more mechanistic approaches like Exact Match (Zou et al., 2023), looking for specific word overlaps. We now explore the potential of PROMPTQUINE on generation tasks, focusing on style transfer, which relies on imperfect quantitative measures, and jailbreaking, which depends on sparse qualitative feedback.

### 4.2.1. TEXT STYLE TRANSFER

**Experimental Setups.** We evaluate our prompts on Yelp sentiment transfer dataset (Shen et al., 2017), as Pilot Study (Appendix B) for both transfer directions. We set aside 200 samples from the original development set for validation and report results on the test set. We use *Joint Score* (Krishna et al., 2020) following and reusing the modules by Deng et al. (2022), as both the fitness function for prompt search and the final metric for performance assessment. We search under the unsupervised setting, where we only need unpaired input samples from original development set (e.g., 100) for fitness estimation, use the entire 200 samples for re-ranking and report final results on the test set. To reduce computations, we adopt an early-stopping strategy inspired by Jamieson & Talwalkar (2016); Li et al. (2018b) to optimize resource allocation. We then compare against

RLPrompt & Promptbreeder under both greedy decoding and Best-of-N sampling (setup of Deng et al. (2022)). To ensure relatively fair comparisons, all methods are given access to the same set of samples (Appendix D.4).

*Table 2.* Automatic evaluation of Yelp Sentiment Transfer. We report their average *Joint Score*, averaging across negative and positive transfer results. The results (no parentheses) are reported with greedy decoding. BoN refers to Best-of-N sampling, following the setups of Deng et al. (2022), i.e., Bo32 under top-10 sampling. Additional results are presented in Table 12 in Appendix D.4.

| Method | GPT-2 | GPT-2 (BoN) | Llama3-8B-It | Llama3-8B-It (BoN) |
|---|---|---|---|---|
| ICL | 4.6 | 40.8 | 54.4 | 69.6 |
| RLPrompt | 10.4 | 51.0 | 4.1 | 54.4 |
| Promptbreeder | 10.2 | 45.3 | 59.1 | 71.8 |
| *TAPruning* | 30.2 | 54.6 | 59.6 | 68.1 |
| PROMPTQUINE | **33.3** | **57.9** | **61.0** | **72.1** |

**Results.** As shown in Table 2, it's possible that PROMPTQUINE could surpass the previous state-of-the-art on this task. These results, as we demonstrate, are largely insensitive to the decoding methods we employed. Additionally, as shown in Table 14 (Appendix D.4), our search framework operates more efficiently, delivering faster search. Nevertheless, compared to few-shot classification tasks, the search tends to be slower due to the imperfect feedback we receive, requiring more samples for fitness estimation.

*Table 3.* Attack success rate (ASR) for jailbreaking comparison of PROMPTQUINE, and conventional ICL. The text in parentheses refers to the fitness measure we used for PROMPTQUINE.

| Attack Method | ASR-EM ↑ | ASR-LLM ↑ |
|---|---|---|
| LLM: `Vicuna-7b-v1.5` | | |
| ICL (2-shot) | 50.4 | 54.7 |
| PROMPTQUINE (ASR-EM) | 99.3 | 97.4 |
| PROMPTQUINE (ASR-LLM) | 99.4 | 97.5 |
| PROMPTQUINE (SV) | 90.3 | 94.2 |
| LLM: `Mistral-7B-Instruct-v0.3` | | |
| ICL (2-shot) | 48.0 | 47.2 |
| PROMPTQUINE (ASR-EM) | 98.8 | 93.8 |
| PROMPTQUINE (ASR-LLM) | 99.8 | 98.1 |
| PROMPTQUINE (SV) | 98.8 | 92.6 |

### 4.2.2. JAILBREAKING

**Evaluation Settings.** We consider this task as highly challenging due to the only sparse, qualitative feedback available, such as the Exact Match score. We adopt a simple few-shot priming setup where the model directly follows the demonstrations to make predictions (i.e., *prefix attack*, without chat templates inserted), using popular models including Vicuna-7b-1.5 (Zheng et al., 2023) and Mistral-7B-Instruct-v0.3 (Jiang et al., 2023a) on AdvBench (Zou et al., 2023). We measure Attack Success Rate (ASR) using both Exact

Match (ASR-EM) and LLM-as-a-Judge by Llama-Guard-3 (Inan et al., 2023) (ASR-LLM). Please refer to Appendix D.4 for the details of our EM strings (Zhao et al., 2024) and prompts for Llama-Guard-3 (Ball et al., 2024) (taken from JailbreakBench (Chao et al., 2024)). We split the original 520 samples into 100 for validation and 420 for testing, using samples from the validation set for fitness estimation and prompt selection. We construct the ICL prompts (2-shot) by reorganizing the 5-shot ICL prompts in Liu et al. (2024).

**Experimental Details.** We produce separated experiments, using both ASR-EM score and ASR-LLM score on the 50 validation samples to guide the search. These experiments are conducted under a purely black-box setting, with the same validation re-ranking for prompt selection. *We also explore to construct more expressive proxies (i.e., to reduce computational budgets) by leveraging steering vectors (SV) from mechanistic interpretability* (Bartoszcze et al., 2025). Such vectors, extracted as activation difference, can enable direct intervention in a model's inner layers to enhance task performance (Rimsky et al., 2024), allowing us to hypothesize that the similarity between internal representation changes after prompting and task steering vectors correlates with performance. Please refer to Appendix D.4 for the full details (3 seeds). We also explore variants of few-shot attack, e.g., in-context attacks (Wei et al., 2023b) where input-output pairs are separated in the conversational contexts, i.e., separated by the chat template tags, in the Appendix D.4.1, where we observe both successes and failures with the current pruning formulations.

**Results.** As shown in Table 3, under the priming setup, PROMPTQUINE is able to derive effective pruning which leads to improved attack results, nearly doubling the traditional average ICL performance. Search results guided by ASR-LLM achieve highest performance. It is noted that SV's performance is generally lower. This is expected as just an initial exploration, and further research is needed to obtain further improvements. We also release some direct predictions of our pruned prompts on the AdvBench in the GitHub repository[1], ensuring rigor given the different data-separation schemes used across prior studies (Jiang et al., 2024; Paulus et al., 2024).

## 5. A Deeper Look at Pruning Effects on ICL

### 5.1. On the Limitations of PromptQuine

While pruning tokens are effective at enhancing overall ICL results, we identify its inherent limitations for current PROMPTQUINE. Specifically, we find that token pruning is not a universally reliable method for stabilizing ICL performance, as its effectiveness remains highly sensitive to

---

[1] github.com/jianyu-cs/PromptQuine/examples/jailbreaking/

| Task | Task Metric Score | | Standard |
|------|:---:|:---:|:---:|
| | $\hat{\Delta}$ (Score($p^{min}$) $\rightarrow$ Score($p^{max}$)) | | Deviation |
| SNLI | 14.3 | $(60.8 \rightarrow 75.1)$ | 7.2 |
| PIQA | 2.6 | $(79.5 \rightarrow 82.1)$ | 1.1 |
| Yelp Positive. | 2.5 | $(49.9 \rightarrow 52.4)$ | 1.4 |
| Yelp Negative. | 1.0 | $(69.8 \rightarrow 70.8)$ | 0.9 |

*Table 4.* Performance fluctuations of PROMPTQUINE-pruned prompts across three random ICL templates, each with three 1-shot ICL seeds. Task scores are reported on the official test set. $\hat{\Delta}$ indicates the performance fluctuation from the minimum to the maximum results across the selected templates.

the chosen ICL templates. We present one such study: we cover a set of tasks, SNLI (Bowman et al., 2015) for classification, PIQA (Bisk et al., 2020) for multi-choice question answering, and Yelp positive transfer and negative transfer (Shen et al., 2017) for generation. For each task, we evaluate two additional ICL templates (see Appendix, Table 24 for examples, where the templates differ only in signal words, separators, spacing characters, and minor variations in natural language instructions as normal variations in practice) and run PROMPTQUINE pruning on each using three different seeds (i.e., three ICL prompts).

As presented in Table 4, differences in templates can still lead to significant variations in task outcomes. For example, in SNLI, the absolute accuracy fluctuation can reach up to 14.3%, which is surprisingly high. This emphasizes that PROMPTQUINE, or fixed-order ICL pruning in general, exhibit instability when exposed to template differences. Incorporating richer, more diverse prompt variations (e.g., token replacement or insertion) can be crucial for further improvement. Nevertheless, achieving this in the current PROMPTQUINE could be hard due to fitness-based selection pressures—beneficial changes, such as the introduction of new tokens, may be prematurely discarded. As a result, we can only adopt more conservative mutation operators, such as varying instructions (Fernando et al., 2024). More aggressive approaches, like exploring the full token space (Deng et al., 2022), are riskier. A promising direction to advance is to consider novelty search (Lehman & Stanley, 2011), which favors novel solutions over fitness alone.

### 5.2. Mechanistic Analysis of Label Words

It is intriguing to analyze what matters in the unconventional pruned ICL prompts for their unreasonable effectiveness. In this section, we examine the role of label words in the demonstrations—aligned with verbalizers—in classification tasks, in line with existing ICL research (Min et al., 2022; Yoo et al., 2022; Wang et al., 2023; Wei et al., 2023a). Concretely, it is often assumed that label words play a crucial

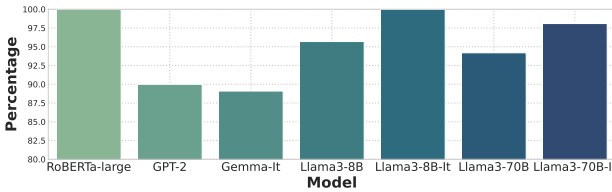

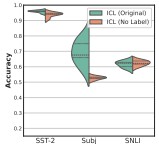 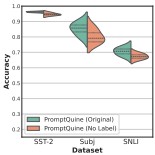 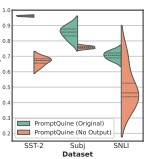

*Figure 3.* The percentage of label word presence is surprisingly high for our PROMPTQUINE-pruned ICL prompts. We obtain these parsing results by exactly matching the label words within the pruned prompts. We verify that the ICL prompts used in the analysis do not contain these words in their exemplar inputs.

*Figure 4.* Changes in (unpruned & pruned) prompting performance on Meta-Llama-3-8B-Instruct when labels are removed (*left & middle*) or when the entire output is removed (*right*, breaking the input-label pairing format). Another study with similar findings on GPT-2 is presented in the Figure 9 in Appendix.

role in determining original ICL performance (Min et al., 2022; Wang et al., 2023). We conduct a series of intervention experiments on label words to better understand their impact on pruning outcomes.

First, we observe that many of the pruned ICL prompts retain certain task-specific label words in their exemplar contexts (Figure 3). This observation closely mirrors findings from conventional ICL, suggesting that label words may play a significant role in ICL, even under pruning. Then, we perform the interventions. Figure 4 (*left*) shows the performance changes after removing the label words in original ICL prompts. The results are consistent with Min et al. (2022) that knowing the label space may help in conventional ICL. The hypothesis also generalizes in our unconventional ICL pruning context (Figure 4, *middle*) that we can observe a slight performance drop on average in pruned prompting performance. We then further experiment with removing the whole output, i.e., the signal words (e.g., `Sentiment:`) and the label words (e.g., `great`) in Figure 4 (*right*). This operation further largely degrades performance, highlighting the importance of preserving input-label format as in standard ICL (Min et al., 2022).

Although most of our findings so far are consistent with the findings on conventional ICL, there are still some special cases where prompt instances violate the aggregated findings discussed above (e.g., SNLI in Figure 4 (*right*), in which the pruned prompt (No Output) could achieve improved performance). This underscores the nuanced sensitivity of prompts within their specific contexts. Finally, we perform experiments with random verbalizers (label words are also changed in prompts). As shown in Appendix, Table 25, all these models show near chance-level performance without pruning. However, we find that counter-intuitively, pruning could also bring some performance improvements, especially towards the large 70B model. A non-negligible number of prompts can achieve significantly improved performance through pruning, and even achieve nearly identical to that of prompts with task-intuitive verbalizers. This is indeed surprising, and we hope future research will explore its underlying mechanisms further.

## 6. Related Work

We discuss briefly various prompt design paradigms and ICL studies in previous work, and provide more comprehensive discussion in Appendix A. Concretely, ICL was introduced by Brown et al. (2020), who demonstrated that LLMs can adapt to downstream tasks using few-shot prompting. Nevertheless, this paradigm has proven to be highly unstable; even slight variations can lead to significant differences in performance. Rubin et al. (2022) demonstrate the importance of demonstration selections, where retrieving demonstrations with similar patterns as the task input typically yields improved performance. Lu et al. (2022) further identifies that even demonstration orders affect the results to a significant extent. Although heuristics—such as retrieving structurally and potentially semantically similar exemplars—often perform well in practice, there remains a lack of well-grounded principles to guide demonstration design. Several studies aim to optimize prompts in alternative formats, typically via direct natural language instructions (Zhou et al., 2022; Guo et al., 2023, *inter alia*). Unlike ICL stabilization, this approach treats prompt optimization as a search for optimal instructions—often using LLMs as prompt engineers. In contrast, we focus on optimizing in an unnatural language space, building directly on few-shot prompts.

## 7. Conclusions

We introduce a novel prompt design paradigm that challenges conventional practices: instead of carefully crafting instructions and examples, we show that pruning random demonstrations into incoherent "gibberish" can still achieve near state-of-the-art performance. As attribution and compression methods remain unreliable, we present PROMPTQUINE, an evolutionary framework that autonomously discovers effective pruning strategies. Experiments across diverse tasks and models validate its effectiveness and runtime efficiency, paving the way for future research into the mechanistic foundations of in-context learning.

## Acknowledgements

We would like to thank Zhen Wang, Yichi Yang, Zhiting Hu for their early discussions; Jiayuan Mao and Freda Shi for their valuable feedback; Yu Rong and the entire DAMO Academy (Hupan Laboratory) for the support.

## Impact Statement

This work aimed at advancing the research of LLMs, with a particular focus on prompt optimization and in-context learning. In contrast to the traditional approach of optimizing in natural language space, we introduce a novel prompt design paradigm that prunes random few-shot prompts into syntactically and semantically unnatural language. Surprisingly, we find that a simple pruning operation applied to ICL prompts is sufficient to match the performance of previous optimization methods across various tasks and models. This insight paves the way for future research, leaving several interpretability questions open for further exploration. For example, it is intriguing to explore why pruning to unnatural language is effective for prompt optimization and why such unnatural ICL remains effective for LLMs. This is particularly noteworthy in cases where pruning enhances performance for several random label words, whereas natural ICL yields only chance-level results.

Moreover, we highlight the direct societal implications of our findings on unnatural language. Notably, our work exposes critical weaknesses in current LLM alignment techniques. Despite extensive training designed to align models with human values and ethical standards when given natural language instructions, our findings reveal that unnatural language can still be used to elicit malicious behaviors—exploiting gaps that developers cannot fully anticipate. As demonstrated in our paper, this vulnerability persists even in large models subjected to extensive red teaming. While continuously iterating on red teaming and eliminating failure cases is beneficial, we advocate for exploring novel alignment techniques that go beyond surface-level fixes. In particular, a stronger focus on *inner alignment* may lead to more robust improvements. For commercial models, we strongly recommend complementing red teaming with output-level restrictions, as this may provide a more intuitive and effective safeguard—especially given that existing alignment methods are primarily optimized for handling natural language inputs.

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

# A. Related Work

**LLM Prompt Optimization.**    LLMs are sensitive to even minor variations in prompts, making their responses difficult to predict without trial and error (Liu et al., 2023). This highlights the need for automated prompt tuning to optimize prompts for specific tasks. Existing research can be broadly classified into two categories: 1) Soft Prompt Tuning (Lester et al., 2021; Qin & Eisner, 2021; Li & Liang, 2021), which optimizes continuous embeddings in the LLM's representation space, replacing discrete prompts with learnable tokens. These embeddings, typically optimized via gradient descent, can outperform hard prompts when applied to intermediate model layers (Sun et al., 2022), similar to parameter-efficient tuning like adapters (Wang et al., 2021); 2) Hard Prompt Optimization: Shin et al. (2020); Deng et al. (2022); Zou et al. (2023); Jones et al. (2023); Choi et al. (2024); Wen et al. (2024) use token-level search algorithms to reward effective tokens and penalize ineffective ones. While promising, these methods often produce non-human-like prompts. In contrast, a more popular line optimizes prompts in natural language, often using LLMs directly as prompt engineers (Zhou et al., 2022; Yang et al., 2023; Guo et al., 2023; Chen et al., 2023; Fernando et al., 2024; Lin et al., 2024; Cui et al., 2024). By combining LLMs with algorithmic designs like evolutionary algorithms (Fernando et al., 2024), these methods achieve expert-level performance.

**In-context Learning Studies.**    Here, we briefly review the literature highlighting the mechanistic studies of ICL, the emergent capability of LLMs. Notably, the underlying mechanisms behind emergent in-context learning remain unclear. The main debate revolves around the mesa-optimization hypothesis (Dai et al., 2023; Von Oswald et al., 2023b;a; Ahn et al., 2023; Cheng et al., 2023; Fu et al., 2023a). Specifically, they argue that these models implement subsidiary learning algorithms that adjust the model inner representations as new inputs are received, with update rules resembling gradient-based optimization of a principled objective. For instance, Von Oswald et al. (2023a) shows that linear self-attention can emulate gradient descent on simple linear regression tasks. However, most of these findings cannot generalize to practical LLM tasks: First, Min et al. (2022) shows that input-label correspondence in demonstrations has little impact on task results. Furthermore, Shen et al. (2023a); Deutch et al. (2024) provide concrete evidence against the validity of such hypotheses in NLP tasks.

**AI alignment.**    Alignment (Gabriel, 2020; Ngo et al., 2022) describes a process of encoding human values and goals into AI assistants to make them as helpful, safe, and reliable as possible. Recent advances of LLMs also motivate extensive studies to align LLM chatbots to human values (Shen et al., 2023b). LLM alignment typically involves two steps. In the instruction-tuning stage (Longpre et al., 2023; Zhang et al., 2023), LLMs are given instruction-response pairs of the tasks so they can learn by imitating the output. In the critique phase, a human or another AI interacts with the model and grades its responses in real-time, known as the reinforcement learning from human or AI feedback (Ouyang et al., 2022; Lee et al., 2023). It seems that LLM researchers are already able to significantly improve the alignment progress as the increasing performance numbers in a wide range of benchmarks and the improved win rates of their optimized LLMs against other LLMs. However, just as Ngo et al. (2022); Di Langosco et al. (2022) hypothesize that, these "aligned" LLMs may only experience superficial outer alignment as they can still be prompted to generate undesired output (Zou et al., 2023; Greenblatt et al., 2024). This necessitates further studies on the inner alignment problem (Ngo et al., 2022).

# B. A Pilot Study with TAPruning

In this section, we provide a pilot study of the *Partial Context Hypothesis* across prompts, models and tasks. As introduced in Section 3.1, where we lack well-established methods for this hypothesis, we begin our investigation with a simple, easy-to-implement hill-climbing search, *TAPruning*. Please refer to Section 3.4 for a more effective algorithmic framework.

## B.1. TAPruning Additional Details

**Algorithmic Descriptions.**    We present the pseudocode of our *TAPruning* algorithm in Algorithm 1. As stated, our algorithm maintains a tracked prompt, denoted as $T$, which serves as the basis for conducting local search operations. As presented in the do-while loop in Algorithm 1, we employ a left-to-right token pruning process, iterating sequentially from the leftmost token to the rightmost, attempting to remove each token at every iteration. The generated new prompt $P$ experiences performance evaluation and comparison against the optimal performance $f^{\text{Optimal}}$. If it improves or at least, degrades within a certain threshold $\delta$, we will accept the prompt $P$ as the new tracked prompt $T$, which we may revisit in the next iteration loop. If it improves, we will also update the optimal prompt $S = P$. Until no further tokens are removable under the performance constraints, we return the optimal prompt $S$ we discovered as the final solution. Please note that

---

**Algorithm 1** Threshold Accepting (*TAPruning*) for Prompt Pruning

---

**Input:** prompt $X$, $x_1 \cdots x_N$, dataset $\mathcal{D}$, performance measure function $f$, language model $\mathcal{M}$, Threshold $\delta$
**Output:** Solution $S$
Measure the performance $f^X$ for prompt $X$ on $\mathcal{D}$ for language model $\mathcal{M}$ as the original prompting performance $f^{\text{Original}}$, which is also used to initialize current optimal prompting performance $f^{\text{Optimal}} = f^{\text{Original}}$.
Initialize tracked prompt $T = X, t_1 \cdots t_N = x_1 \cdots x_N$.
Initialize optimal prompt $S = X, s_1 \cdots s_N = x_1 \cdots x_N$.
**repeat**
    Initialize length variable $L$ as the number of the tokens in current tracked prompt $T$ (re-tokenized).
    **for** $i = 1$ **to** $L$ **do**
        Generate a neighborhood prompt $P$ by removing the token $t_i$ from prompt $T$.
        Calculate the performance $f^P$ for $P$ on $\mathcal{D}$,
        If $f^P > f^{\text{Optimal}}$, we accept and update the prompt $T = P$, optimal prompt $S = P$, and update the optimal performance record $f^{\text{Optimal}} = f^P$.
        If $f^P < f^{\text{Optimal}}$, we accept $P$ as the new $T$ only if $f^P > f^{\text{Optimal}} \times \delta$ (within the threshold).
    **end for**
**until** The solution $S$ converges

---

**Algorithm 2** Steepest-Ascent Hill Climbing (*SAHCPruning*) for Prompt Pruning

---

**Input:** prompt $X$, $x_1 \cdots x_N$, dataset $\mathcal{D}$, performance measure function $f$, language model $\mathcal{M}$, Threshold $\delta$
**Output:** Solution $S$
Measure the performance $f^X$ for prompt $X$ on $\mathcal{D}$ for language model $\mathcal{M}$ as the original prompting performance $f^{\text{Original}}$, which is also used to initialize current optimal prompting performance $f^{\text{Optimal}} = f^{\text{Original}}$.
Initialize tracked prompt (also as optimal prompt) $S = X, s_1 \cdots s_N = x_1 \cdots x_N$.
Initialize optimal token index candidate $T_c = -1$.
**repeat**
    Initialize length variable $L$ as the number of the tokens in current tracked prompt $S$ (re-tokenized).
    **for** $i = 1$ **to** $L$ **do**
        Generate a neighborhood prompt $P$ by removing the token $s_i$ from prompt $S$.
        Calculate the performance $f^P$ for $P$ on $\mathcal{D}$,
        If $f^P > f^{\text{Optimal}}$, update the optimal performance record $f^{\text{Optimal}} = f^P$ and optimal token index candidate $T_c = i$.
    **end for**
    **if** $T_c$ is not -1 **then**
        Update tracked prompt $S$ by removing token $s_{T_c}$
        Update $T_c = -1$
    **end if**
**until** The solution $S$ converges

---

the tokens (tokenized) may change through the optimization as the prompt changes in its surface form, as we desire the optimized prompts to be portable strings. Preliminary experiments show that the *Partial Context Hypothesis* also works if we retain the same prompt tokens through optimization.

We discuss other design choices, especially attempting to leverage insights from LLM mechanistic interpretability studies, such as attribution methods (Li et al., 2016a), in Appendix C. These methods can generate a token ranking in a single forward-backward pass, potentially leading to significant speedups. However, as observed, they generally fail to effectively guide performance improvement, making them unsuitable for our problem. We present *SAHCPruning* in Algorithm 2, which accepts the update only if this move (i.e., how to remove the next single token) is the best possible move among all available options. This algorithm limits in its high computational cost: since ICL prompts always take more than hundreds of tokens, i.e., the problem is relatively high-dimensional, the search would become much slower and less scalable, e.g., Table 11. Also, due to the deceptive, multimodal nature of the search landscape, *SAHCPruning* does not always outperform *TAPruning* (Table 1).

*Table 5.* Details of our classification datasets evaluated in this work. $|C|$: # of classes for classification tasks. $|$Test$|$: # of testing samples for each particular dataset.

| Dataset | Type | $|C|$ | $|$Test$|$ | Label words |
|---------|------|-------|------------|-------------|
| SST-2 | Sentiment (Movie reviews) | 2 | 1.8k | terrible, great |
| Subj | Subjectivity (Movie reviews) | 2 | 2k | subjective, objective |
| AG's News | Topic (News articles) | 4 | 7.6k | World, Sports, Business, Tech |
| Yelp-5 | Sentiment (Yelp reviews) | 5 | 50k | terrible, bad, okay, good, great |
| SNLI | Natural Language Inference | 3 | 10k | Yes, Unknown, No |
| Yahoo | Topic (Question types) | 10 | 60k | culture, science, health, education, computer, sports, business, music, family, politics |

## B.2. Pilot Study

**Datasets.** Our investigation covers four task types: classification, multiple-choice question answering, generation, and chain-of-thought reasoning. Unless otherwise stated, we sample from their official validation set for the prompt selection, and report on their official testing split for final evaluation. 1) Classification: We evaluate sentiment analysis (SST-2 (Socher et al., 2013), Yelp-5 (Asghar, 2016)), subjectivity classification, Subj (Pang & Lee, 2004), topic classification (AG's News (Zhang et al., 2015) and Yahoo (Labrou & Finin, 1999)), and natural language inference (SNLI (Bowman et al., 2015)). We present the overall statistics in Table 5; 2) Multi-choice questions: We include commonsense reasoning datasets, PIQA (Bisk et al., 2020), a binary-choice question answering dataset with verbalizers A and B. Since its testing performance can only be assessed via submission to the leaderboard, we sample from their training set to form our held-out set for prompt selection and evaluate on the official validation set, which consists of 2,000 examples, for the final performance assessment; 3) Generation: We include Yelp Sentiment Transfer (Shen et al., 2017) (Yelp Style.), where we follow Deng et al. (2022) in an unsupervised style transfer setting, sampling from its development set for prompt selection and using its test set for final evaluation, with reference collected by Li et al. (2018a) for both transfer directions (e.g., positive-to-negative and negative-to-positive); 4) Chain-of-Thought (CoT) reasoning: We include GSM8K (Cobbe et al., 2021) and MAWPS (Koncel-Kedziorski et al., 2016).

**Models and Baselines.** We evaluate a range of popular LLMs varying in architecture, scale, and alignment efforts. Concretely, 1) Classification: we study base models: RoBERTa-large (Liu et al., 2019), GPT-2 (Radford et al., 2019), Meta-Llama-3-8B, Meta-Llama-3-70B (AI@Meta, 2024), instruction-tuned model (SFT): Gemma-7b-it (Team et al., 2024), and further reinforcement learning-tuned (RLHF) models: Meta-Llama-3-8B-Instruct, and Meta-Llama-3-70B-Instruct (AI@Meta, 2024); 2) Multi-choice question answering: Meta-Llama-3-8B & Meta-Llama-3-8B-Instruct; 3) Generation: GPT-2 & Meta-Llama-3-8B-Instruct; 4) Reasoning: Meta-Llama-3-8B-Instruct, Mistral-7B-Instruct (Jiang et al., 2023a) and Qwen2-7B-Instruct (Yang et al., 2024a). In this section, we focus on results from RoBERTa-large and Meta-Llama-3-8B-Instruct. Results for other models, along with PROMPTQUINE, are presented in the corresponding Appendix sections that follow. We include both the original ICL and RLPrompt (Deng et al., 2022)—a state-of-the-art prompt optimization method—as well as a token-level search algorithm for the "secret language" baseline. It is important to note that this approach cannot be applied to CoT reasoning tasks. For ICL prompts, we primarily use one-shot ICL, where each prompt for four-way classification is constructed by randomly sampling one instance and its label from each category in the training split. Two-shot prompts (two input-output pairs) are used for style transfer. For ICL on classification and multiple-choice question answering, results are averaged over 10 random seeds, each with a unique ICL prompt. For generation and reasoning tasks, we average over 5 seeds. For RLPrompt, results are averaged over 3 seeds. We also use the entire held-out set for prompt selection in RLPrompt. Specifically, we first rank the explored prompts by their reward scores, select the top 50, and then re-rank them based on validation performance. We also explore using more training task samples for reward calculation but observe minimal performance gains. Thus, we adopt the original 16-shot setup from Deng et al. (2022) for classification and use 200 unsupervised samples for style transfer, training over five tokens with default hyperparameters.

**Prompt Template Details.** First, for the RLPrompt, we use the following template for most of the classification tasks, following (Schick & Schütze, 2021; Deng et al., 2022):

*RLPrompt - Classification:*

```
{Input}{Prompt}
```

That is, we replace the "{Input}" placeholder with the task input instance. The policy network directly generates the prompt, replacing the "{Prompt}" placeholder. For masked language models, such as RoBERTa-large, we directly append a `[MASK]` into the last of the template, run the prompt, and parse its task predictions from that token. One exception is the natural language inference task, which involves two inputs (the premise & the hypothesis). We provide its template as below:

*RLPrompt - Natural Language Inference:*

```
{Premise} {Hypothesis} {Prompt} Entailment:
```

Then, we provide its template for multi-choice question answering:

*RLPrompt - Multi-choice Question Answering:*

```
{Prompt}
Question: {Input}
Options: A) {Option1} B) {Option2}
Answer:
```

The template for style transfer:

*RLPrompt - Text Style Transfer:*

```
{Prompt} "{Input}" "
```

Just as (Deng et al., 2022) does, we allow the model to generate tokens one by one until we meet a special character `"` (the quotation mark). We then parse the completions inside the quotation marks as final task output.

Next, we provide our templates for our ICL prompts, which also serve as the basis for our prompt pruning.

*ICL - Sentiment Analysis (SST-2 and Yelp-5):*

```
{Examples}

Review: {Input}
Sentiment:
```

For "{Examples}", we use the same format to organize the input-output task pairs as we do for the input task instances. This principle applies to all our ICL prompts regardless of tasks.

We discuss templates for other tasks as follows:

*ICL - Subjectivity Classification (Subj):*

```
{Examples}

Sentence: {Input}
Viewpoint:
```

*ICL - News Topic Classification (AG's News):*

```
{Examples}

Article: {Input}
Answer:
```

*ICL - Natural Language Inference (SNLI):*

```
{Examples}

Hypothesis: {Hypothesis}
Premise: {Premise}
Given the premise, is the hypothesis true? Yes, No or Unknown?
The answer is:
```

*ICL - Topic Classification (Yahoo):*

```
{Examples}

Sentence: {Input}
Topic:
```

*ICL - Multi-choice Question Answering (PIQA):*

```
{Examples}

Question: {Input}
Options: A) {Option1} B) {Option2}
Answer:
```

*ICL - Text Style Transfer (Yelp Sentiment.):*

```
{Examples}

Here is a text, which is [negative/positive]: "{Input}".
Here is a rewrite of the text, which is [positive/negative]: "
```

We alternate between the sentiment signal words, "negative" and "positive", for two token choices for target sentiment direction. For example, for positive transfer, we use negative and positive respectively. Similarly, we parse the completions inside the quotation marks as final task output.

*ICL - Chain-of-thought Reasoning (GSM8K & MAWPS):*

```
{Examples}

Question: {Input}
Let's think step by step.
```

The models are then expected to follow the exemplar patterns, i.e., generating the reasoning chains before predicting the task answer. For this particular task, as we lack the direct annotations of the ground truth reasoning chains, we conduct the following steps to collect our CoT prompts: 1) GSM8K (Cobbe et al., 2021): we directly sample and adopt the ICL prompts provided by the Chain-of-Thought Hub (Fu et al., 2023b); 2) MAWPS (Koncel-Kedziorski et al., 2016): we ask GPT-4o (Hurst et al., 2024) to generate the reasoning chains for the training questions we sampled, which are then paired to construct the corresponding ICL prompts.

**Evaluation settings.** During the search stage, each prompt's quality is evaluated on 200 samples from the official validation split (as our held-out set), or the training split if the validation is unavailable (e.g., PIQA). We report performance on the official test set (i.e., validation set for PIQA).

**Evaluation Metrics.** As stated, we report testing accuracy for classification, multi-choice question answering and reasoning tasks. For style transfer, we follow Deng et al. (2022) to use their fine-tuned style classifiers for *Style* calculation, pre-trained LM to calculate input-output alignment (Deng et al., 2021), *Content*, and a pre-trained grammaticality classifier (Krishna et al., 2020) for *Fluency*. Then, we average these sentence-level scores, as the *Joint Score*, strictly following

*Table 6.* Task performance evaluation of *TAPruning* upon original 1-shot ICL prompts. For classification, multi-choice answering (PIQA) and math reasoning (GSM8K & MAWPS), we report *accuracy* on their test set. For style transfer (Yelp sentiment transfer, Yelp Style.), we report their *joint score* on a separate 500 sample set, following (Krishna et al., 2020; Deng et al., 2022). We evaluate using greedy decoding. We report classification performance on RoBERTa-large (Liu et al., 2019) and overall task performance on Meta-Llama-3-8B-Instruct (AI@Meta, 2024) (Llama3-8B-It). Note that RLPrompt (Deng et al., 2022) cannot be applied to CoT reasoning, so we leave those cells blank. The numbers in (parentheses) are the standard deviations between different prompts.

| LLM | Methods | SST-2 | Subj | AG's News | Yelp-5 | SNLI | Yahoo | Yelp Style. | PIQA | GSM8K | MAWPS |
|---|---|---|---|---|---|---|---|---|---|---|---|
| | ICL | 86.2 (7.7) | 53.8 (5.3) | 57.2 (3.6) | 27.3 (2.1) | 33.3 (1.2) | 36.8 (9.9) | - | - | - | - |
| RoBERTa-large | RLPrompt | 92.5 (0.8) | 81.2 (1.7) | 80.2 (0.7) | 44.8 (4.3) | 33.5 (0.8) | 48.6 (1.0) | - | - | - | - |
| | *TAPruning* (Ours) | 90.8 (2.6) | 80.9 (1.6) | 79.7 (2.3) | 42.8 (6.9) | 42.0 (5.9) | 51.4 (1.5) | - | - | - | - |
| | ICL | 95.9 (0.6) | 66.7 (4.3) | 83.7 (1.9) | 52.2 (6.0) | 61.9 (2.0) | 57.1 (6.9) | 54.4 (6.9) | 75.4 (1.6) | 68.0 (7.4) | 75.6 (9.4) |
| Llama3-8B-It | RLPrompt | 88.4 (1.5) | 82.9 (0.5) | 84.7 (0.7) | 48.1 (1.0) | 42.4 (4.2) | 58.5 (0.6) | 8.3 (2.8) | 85.9 (2.6) | - | - |
| | *TAPruning* (Ours) | 95.0 (1.5) | 74.5 (3.9) | 88.6 (0.3) | 60.2 (0.9) | 68.6 (2.9) | 61.7 (1.7) | 59.6 (1.3) | 75.1 (3.0) | 77.1 (2.1) | 85.0 (3.9) |

Krishna et al. (2020)'s protocol:

$$J(\text{Content}, \text{Style}, \text{Fluency}) = \tag{2}$$
$$\text{mean}_{\mathbf{x} \in \mathcal{X}} \left( \text{Content}(\mathbf{x}) \cdot \text{Style}(\mathbf{x}) \cdot \text{Fluency}(\mathbf{x}) \right).$$

**Results.** As shown in Table 6, pruning the original ICL prompts can improve performance, supporting our *Partial Context Hypothesis*. Surprisingly, this approach works well for chain-of-thought reasoning, where the outputs are highly structured and answers are distant from the prompt—an unexpected result for such a complex task. Remarkably, pruned prompts could deliver competitive performance, often surpassing RLPrompt across several tasks, effectively bridging the gap between the "secret language" (i.e., unnatural language artifacts discovered by previous token-level search algorithms) and original natural language prompts. Note that the improvement is independent of the ICL prompts we sampled, suggesting a potentially more effective approach to stabilize ICL performance (Lu et al., 2022; Rubin et al., 2022). We believe these results can inspire novel prompt optimization or in-context learning stabilization algorithms. We thus explore this further in Section 3.4, with our *TAPruning* serving as a baseline for the pruning-based prompt optimization. Despite its simplicity, this baseline remains competitive—being comparable or directly outperforming previous state-of-the-art methods in both performance and efficiency (e.g., Table 11), making it a strong contender. It is also intriguing to analyze what is left after the pruning, which can potentially inspire some mechanistic insights. We provide such analysis in Section 5.2.

*Table 7.* Zero-shot chain-of-thought performance on the MultiArith (Roy & Roth, 2015) dataset using InstructGPT (text-davinci-002) (Ouyang et al., 2022). The natural language prompt was proposed in (Zhou et al., 2022) to enable the zero-shot chain-of-thought reasoning of large language models. We produce these experiments using the answer extraction script from (Kojima et al., 2022).

| No. | Category | Zero-shot CoT Trigger Prompt | Accuracy |
|---|---|---|---|
| 1 | APE (Zhou et al., 2022) | Let's work **this** out **in a** step by step **way** to be sure we have the right answer. | 81.5 |
| 2 | Pruned | Let's work out step by step **to be** sure we **have the** right answer. | 85.3 |
| 3 | | Let's work out step by step sure we right answer. | **86.7** |
| - | | (Empty) | 17.7 |

# C. Alternative Prompt Pruning Design Choices

In this section, we discuss some potential alternative design choices that may be used in prompt pruning to provide more background about why we start with the hill-climbing method describe in Section B in exploring the *partial Context Hypothesis* and inspire future works.

The algorithms discussed below aim to improve prompt pruning speed by leveraging the information inherently provided by LLMs for the given task. Approaches requiring significant efforts, such as probing techniques to learn task proxies (Zhu et al., 2022), which depend on large collections of prompt-performance pairs for supervision, are not considered due to their impracticality. We also do not include the studies of ICL-gradient descent correspondence hypothesis here, where the

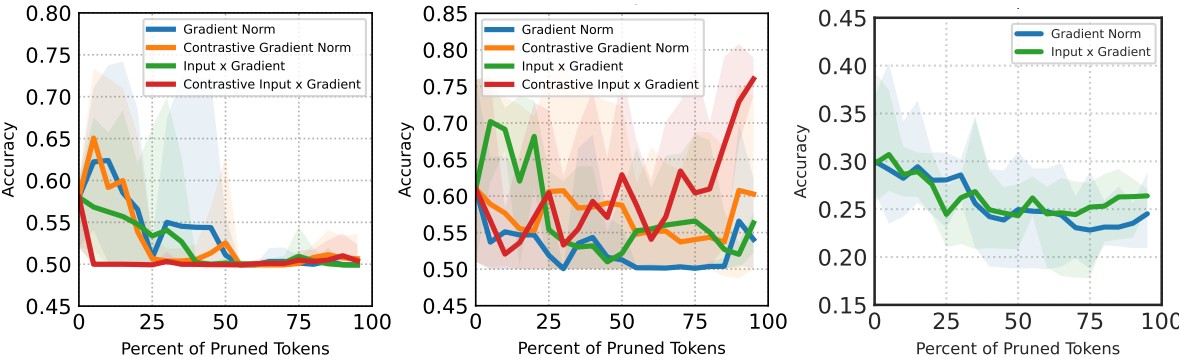

(a) How the Task *Testing Accuracy* changes (SST-2, Subj and AG's News), guided by attribution scores, when increasing the percent of pruned tokens.

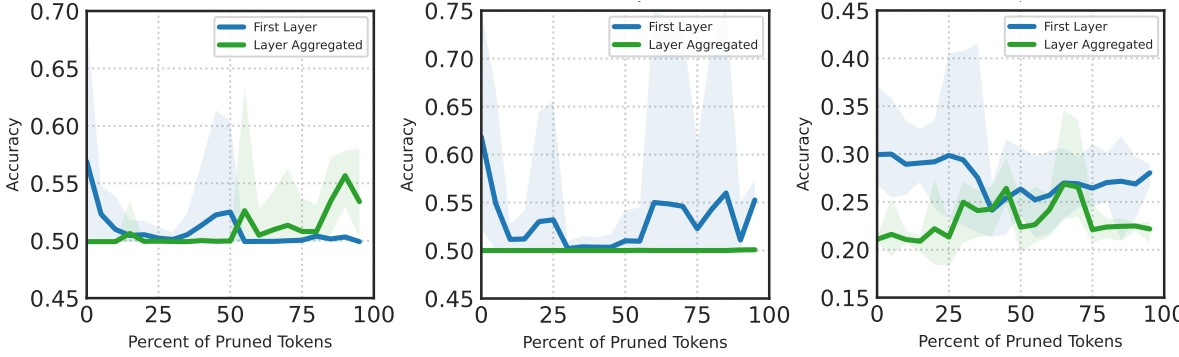

(b) How the Task *Testing Accuracy* changes (SST-2, Subj and AG's News), guided by attention weights, when increasing the percent of pruned tokens.

*Figure 5.* Pruning-based prompting performance using a variety of methods (*Top* for attribution scores guided pruning, and *Bottom* for attention weights guided pruning).

representational similarity may indicate task performance, as recent work (Shen et al., 2023a; Deutch et al., 2024) invalidate the effectiveness of such hypothesis in generalizing to practical NLP tasks for LLMs. Please note that the algorithms below are all inspired by the developments in mechanistic understandings of LLMs. Some of them might only be applicable in white-box scenarios.

**Instance attribution scores as guidance.** Instance attribution methods aim to improve interpretability by identifying influential tokens for a model's prediction. Common approaches include gradient-based methods (e.g., gradient × input saliency (Baehrens et al., 2010; Li et al., 2016a)) and erasure-based methods (Li et al., 2016b; Feng et al., 2018), which measure the change in output when masking tokens. Our approach (e.g., TA) aligns with erasure methods, which are computationally expensive due to the need for multiple input perturbations. This raises the question of whether gradient-based methods can be adapted to approximate token rankings within a single forward-backward pass (Feng et al., 2018). If so, we can directly use the attribution scores to guide the pruning.

**Token attention weights as guidance.** Attention mechanisms are crucial for understanding how LLMs process prompts and assign token importance. Analyzing attention distributions across layers can rank token significance, with higher attention scores indicating greater influence on task prediction (Xiao et al., 2023; Ge et al., 2023). We aggregate attention weights across heads in specific layers to compute token importance scores, guiding pruning.

## C.1. Experiments

We evaluate the aforementioned methods on several classification datasets and demonstrate their limited utility in the context of prompt compression as guided prompt search reformulation.

*Table 8.* Classification accuracies of optimal prompts for *TAPruning* and alternative pruning methods based on attribution scores and attention weights in GPT-2 across various datasets.

| Method | SST-2 | Subj | Yelp-5 |
|---|---|---|---|
| ICL (1-shot, original) | 54.2 (4.8) | 64.3 (9.8) | 30.9 (5.2) |
| Gradient Norm | 66.5 (7.5) | 72.3 (0.0) | 33.6 (4.6) |
| Input $\times$ Gradient | 66.8 (5.2) | 75.1 (1.7) | 33.9 (3.5) |
| Contrastive Gradient Norm | 66.1 (8.6) | 75.9 (2.4) | 31.5 (2.8) |
| Contrastive Input $\times$ Gradient | 52.0 (1.5) | 75.7 (2.9) | 31.5 (2.8) |
| First Layer | 57.3 (7.0) | 74.7 (2.2) | 35.7 (4.8) |
| Layer Aggregated | 60.2 (4.1) | 72.3 (0.0) | 30.2 (3.0) |
| *TAPruning* (Ours) | **68.1 (10.6)** | **75.9 (2.5)** | **39.9 (1.8)** |

**Experimental Setups.**    We perform experiments on SST-2, Subj, and Yelp-5 using GPT-2, sampling five ICL prompts per dataset. Importance scores are computed once based on the same 200 samples, with performance evaluated on the official test split. 1) Instance attribution scores: we experiment with state-of-the-art approaches, including Gradient Norm (Simonyan, 2013; Li et al., 2016a) and Input $\times$ Gradient (Denil et al., 2014; Shrikumar et al., 2016) as well as their contrastive versions (Yin & Neubig, 2022). Token importance scores are aggregated across the 200 samples to generate final rankings, which are then used to guide token pruning; 2) Token attention weights: For each task, we use the first-layer attention scores and the aggregated attention scores (average) across all layers as the *First Layer* and *Layer Aggregated* importance scores, respectively. Prompt pruning is then performed according to the derived attention scores.

**Results.**    We examine the impact of increasing token pruning percentages on task accuracy (Figure 5). If the derived scores guide pruning effectively, accuracy should first increase monotonically, then decrease beyond a certain threshold. However, as shown in Figure 5, all approaches exhibit nonlinear fluctuations, with their optimal performance still lagging behind our hill-climbing approach (Table 8). This suggests these methods are ineffective at guiding pruning. Also, please note that unlike the hill-climbing approach discussed in Section B, the methods presented in this section do not converge to a single optimal prompt. Instead, as shown in Table 8, we can only obtain the results by experimenting with and selecting from all the prompts derived on the validation set based on their importance scores, which is also less flexible. Finally, just as the failures of existing prompt compression methods to guide the pruning (shown in Table 1), we stick with our *TAPruning*, the hill-climbing approach, in section B to first gain a basic intuition of our *Partial Context Hypothesis*.

# D. PROMPTQUINE Additional Details

## D.1. Genetic Algorithm Details

**Introduction.**    As outlined in Algorithm 3, we evolve a population $\mathcal{P}$ of $\#p$ individuals (here, pruned ICL prompts $P$ as *phenotypes*). At every generation, each $P$ is evaluated, producing a fitness score $f(P)$. The selection process then identifies high potential individuals typically the best-performing ones, using methods like tournament selection. Genetic operators, such as mutation, are applied to generate offspring: mutation introduces small, random changes (e.g., pruning tokens), while crossover (i.e., exchanging tokens) is excluded in this case due to its limited benefits on performance and its potential to complicate the search space, which is also a common practice (Real et al., 2017; Co-Reyes et al., 2021). The offspring are then evaluated, and elitism-based selection determines which individuals survive to the next generation. This process repeats iteratively until a termination condition is met, such as achieving a satisfactory fitness level or reaching a predefined number of generations.

**Implementation Details.**    Here, we present the details of our GGA and SSGA implementations discussed in Section 3.4. Specifically, we provide their pseudo-codes: Algorithm 4 for GGA and Algorithm 5 for SSGA, both of which replace the main loop in Algorithm 3. Each approach has distinct trade-offs. As illustrated in Algorithm 4, GGA divides the reproduction process into multiple generations, with several reproduction events occurring within each. In our setup, the number of offspring ($\#c$) often exceeds the population size ($\#p$), encouraging each individual to participate as a parent.

---

**Algorithm 3** Genetic Prompt-Quine (PROMPTQUINE) Framework for Prompt Subsequence Search

---

**Input:** prompt $X$, $x_1 \cdots x_N$, training dataset $\mathcal{D}_{train}$, validation dataset $\mathcal{D}_{val}$, fitness function $f$, primary task performance measure $m$, population size $\#p$, sample size $\#s$, number of iterations $\#n$, language model $\mathcal{M}$, elite calibration selection function $R$

**Output:** Prompt $P$

$\mathcal{P} \leftarrow$ `population-init`$(X, \mathcal{M}, f, \mathcal{D}_{train}, \#p)$.

History $\mathcal{H} \leftarrow \mathcal{P}$.

**for** $i = 1$ **to** $\#n$ **do**

    $\mathcal{S}^{(i)} \leftarrow$ `sample`$(\mathcal{P}, \#s)$.

    Parent $\leftarrow$ `select`$(\mathcal{S}^{(i)})$.

    Child $\leftarrow$ `copy-then-mutate`(Parent).

    Scores $\leftarrow$ `evaluate`(Child, $M, f, \mathcal{D}_{train}$).

    $\mathcal{P}, \mathcal{H} \leftarrow$ `push-pop`(Child, Scores, $\mathcal{P}, \mathcal{H}, \#p, i$).

**end for**

Elite $\mathcal{E} \leftarrow R(\mathcal{H})$

Prompt $P \leftarrow$ `prompt-rank`$(\mathcal{E}, M, m, \mathcal{D}_{valid})$.

---

The framework converges either when a predefined iteration limit is reached or, more commonly, when the minimal prompt length threshold ($\#l$) is met. GGA is well-suited for parallelization, which can further enhance the results by leveraging computations. In our current implementation, we mainly use batching along with efficient LLM serving tools, such as vLLM (Kwon et al., 2023), which demonstrates improved runtime efficiency.

The core difference between SSGA and GGA lies in SSGA's self-adaptation of parameters, such as mutation rate and selection pressure, throughout the evolutionary process. As outlined in Algorithm 5, SSGA continuously updates the population with new offspring at each step. We also allow more offspring before inclusion through regularized evolution, introducing genetic diversity in real time—akin to a self-adaptive mutation rate. Additionally, by dynamically increasing the population size, SSGA creates a self-adaptive selection pressure, potentially increasing the likelihood of each individual becoming a parent. The evolution process terminates under the same conditions as GGA, typically when the iteration limit is reached. Compared to GGA, SSGA is more exploratory but sacrifices some runtime efficiency due to the lack of batching and parallelization. As a result, we primarily use SSGA for 1-shot ICL pruning experiments (10,000 iterations), while GGA, which converges faster, is used for more-shot ICL pruning experiments. Later experiments show that, for most landscapes we encounter, GGA performs comparably to SSGA.

*Table 9.* **Hyperparameters.** Through extensive experiments, the current hyperparameters have proven stable across the tasks presented in the 1-shot ICL pruning studies. For more-shot experiments, we recommend increasing the number of iterations, and, if computational resources permit, enlarging the population and offspring sizes as well.

| Hyperparameter | Default Value | Description |
|---|---|---|
| Population Size | 30 | How many individuals per population. |
| Offspring Size | 50 | How many offspring individuals generated per generation. |
| Mutation rate | [1,2,3,4] | The possible range of bits we change for reproduction. |
| Tournament Selection Ratio | 0.2 | The ratio of individuals sampled for tournament selection. |
| Number of Iterations | 10,000 | The maximal number of prompts we explored through pruning. |
| Minimal Prompt Length Threshold | 15 | The expected minimum average population prompt length for search termination. |

**Hyperparameters.** We present our shared hyperparameters for both SSGA and GGA under 1-shot ICL pruning in Table 9. If additional resources are available, we highly recommend increasing both the population and offspring sizes, as this can yield significant performance benefits. Moreover, we believe that employing a more adaptive mutation rate can be beneficial, which we leave as future work (e.g., using a larger mutation rate in the early stages and gradually annealing it over time may enhance performance).

---

**Algorithm 4** PROMPTQUINE's Generational GA (GGA) implementation for Prompt Subsequence Search

---

**Input:** Initial population $\mathcal{P}$, with population size $\#p$, number of maximal iterations $\#n$, minimal prompt length threshold $\#l$, training dataset $\mathcal{D}_{train}$, fitness function $f$, offspring size $\#c$, language model $\mathcal{M}$, tournament selection ratio $\#k$, mean population prompt length calculation function $h$
**Output:** Prompt History $\mathcal{H}$
History $\mathcal{H} \leftarrow \mathcal{P}$.
**for** $i = 1$ **to** $\#n$ **do**
   Initialize $g \leftarrow Empty$.
   Initialize mean population prompt length $L \leftarrow h(\mathcal{P})$.
   **if** $L < \#l$ **then**
     **break**
   **end if**
   **for** $j = 1$ **to** $\#c$ **do**
     Parent $\leftarrow$ `Tournament-Selection`$(\mathcal{P}, \#k)$.
     Child $\leftarrow$ `copy-then-mutate`(Parent).
     Score $\leftarrow$ `evaluate`(Child, $M, f, \mathcal{D}_{train}$).
     $g, \mathcal{H} \leftarrow$ `push`(Child, Score).
   **end for**
   Sort $g$ in descending order of Score.
   Update population $\mathcal{P} \leftarrow g[: \#p]$.
**end for**

---

**Algorithm 5** PROMPTQUINE's Steady-state GA (SSGA) implementation for Prompt Subsequence Search

---

**Input:** Initial population $\mathcal{P}$, with population size $\#p$, number of maximal iterations $\#n$, minimal prompt length threshold $\#l$, training dataset $\mathcal{D}_{train}$, fitness function $f$, offspring size $\#c$, language model $\mathcal{M}$, tournament selection ratio $\#k$, mean population prompt length calculation function $h$
**Output:** Prompt History $\mathcal{H}$
History $\mathcal{H} \leftarrow \mathcal{P}$.
**for** $i = 1$ **to** $\#n$ **do**
   Initialize mean population prompt length $L \leftarrow h(\mathcal{P})$.
   **if** $L < \#l$ **then**
     **break**
   **end if**
   **for** $j = 1$ **to** $\#c$ **do**
     Parent $\leftarrow$ `Tournament-Selection`$(\mathcal{P}, \#k)$.
     Child $\leftarrow$ `copy-then-mutate`(Parent).
     Score $\leftarrow$ `evaluate`(Child, $M, f, \mathcal{D}_{train}$).
     $\mathcal{P}, \mathcal{H} \leftarrow$ `push`(Child, Score).
   **end for**
   Initialize $g \leftarrow \mathcal{P}[\#p :]$.
   Sort $g$ in descending order of Score.
   Update population $\mathcal{P} \leftarrow g[: \#p]$.
**end for**

---

### D.2. Multimodal Landscape Additional Studies

In the main paper, we present the studies only using instruction-tuned Meta-Llama-3-8B-Instruct. Here, we present additional results for base model, Meta-Llama-3-8B.

The results in Figure 6 show that greedy hill-climbing alone fails to consistently converge to the same prompt solutions, underscoring the multimodal nature of the ICL landscape. An exploratory search approach may prove more effective in improving the results. Additionally, as shown in both middle and right subfigures, RS is inefficient at obtaining high-quality prompts, whereas ES is more effective, potentially leading to better local optima under restricted computational budgets,

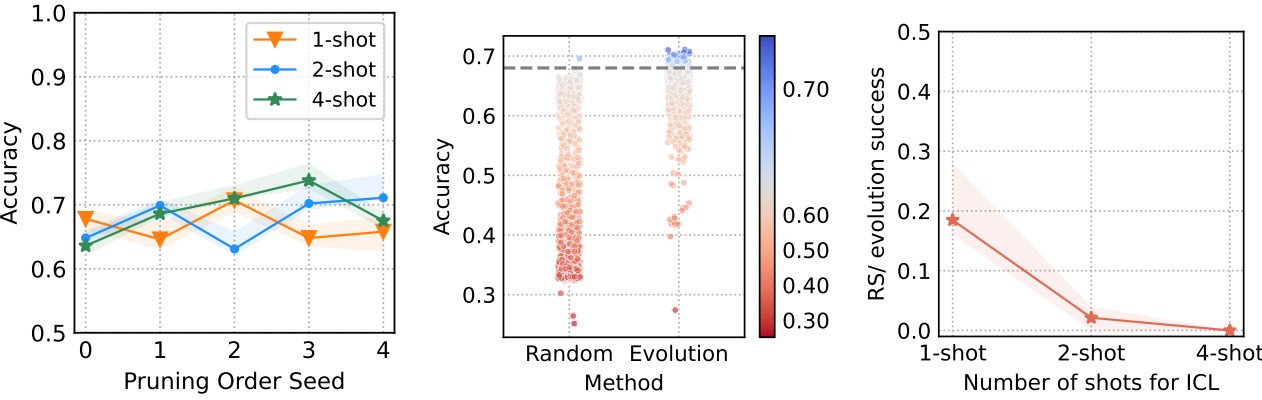

(a) Optimization challenges in our ICL-initialized landscape using Meta-Llama-3-8B-Instruct for natural language inference (SNLI).

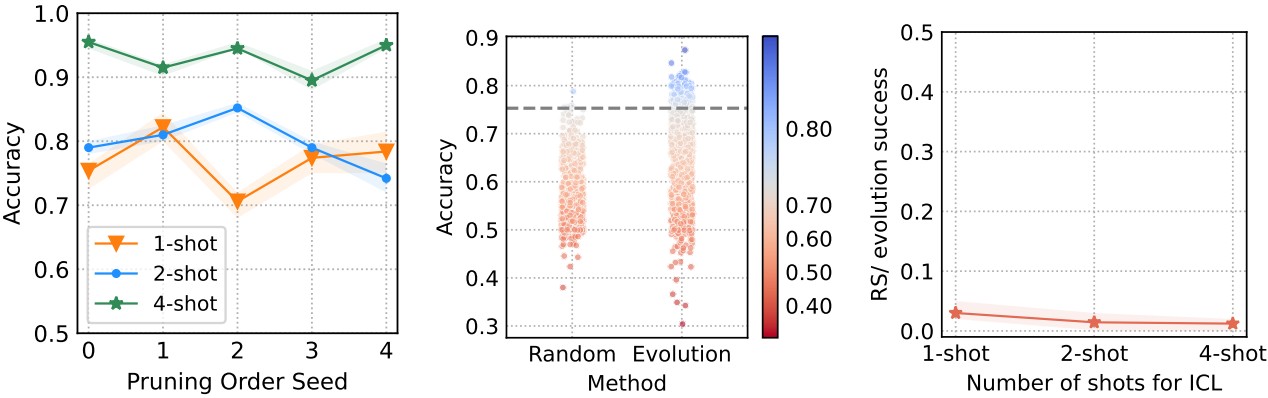

(b) Optimization challenges in our ICL-initialized landscape using Meta-Llama-3-8B for subjectivity classification (Subj).

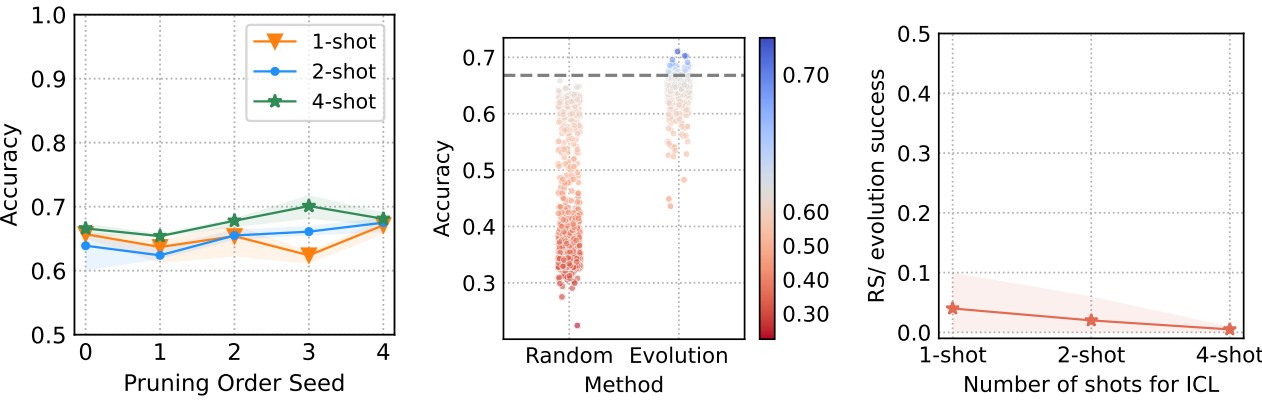

(c) Optimization challenges in our ICL-initialized landscape using Meta-Llama-3-8B for natural language inference (SNLI).

*Figure 6.* Additional results on both Llama-3-8B base and instruct models, revealing the complex, multimodal nature of the ICL search landscape.

further corroborating our findings in the main paper.

## D.3. PROMPTQUINE for Classification

**Fitness Function.**    Our preliminary experiments show that the piecewise reward function proposed in RLPrompt (Deng et al., 2022) is highly effective at distinguishing between prompts of varying quality, compared to a variety of probability-only selection approaches (Yang et al., 2024b). We thus provide its details below, as our classification task proxy:

$$R(\mathbf{z}, \mathbf{x}, c) = \lambda_1^{1-\text{Correct}} \lambda_2^{\text{Correct}} \text{Gap}_\mathbf{z}(c), \tag{3}$$

where the reward is defined as the gap between the label probability and the highest probability from other classes for a given prompt $\mathbf{z}$ and training example $(\mathbf{x}, c)$. The gap is written as $\text{Gap}_\mathbf{z}(c) := P_\mathbf{z}(c) - \max_{c' \neq c} P_\mathbf{z}(c')$, where $P_\mathbf{z}(c) := P_{\text{LM}}(c|\mathbf{z}, \mathbf{x})$ to denote the probability of label $c$. The gap value is positive when the prediction is correct, and negative otherwise. We denote $\text{Correct} := \mathbb{1}[\text{Gap}_\mathbf{z}(c) > 0]$ that for a correct prediction, the positive reward is further multiplied by a larger number to signal its desirability. We set $\lambda_1$ and $\lambda_2$ as 180 and 200, following (Deng et al., 2022).

**Re-ranking Mechanisms.**    As pointed out in the main paper and algorithm 3, we incorporate specific prompt re-ranking mechanisms in later stages. This is because, despite the effectiveness of the fitness function in Equation 3, it can still be exploited due to its imperfect design. This may stem from the limited number of task samples used for fitness estimation or from inherent imperfections and complexities within the fitness function itself. This is inevitable, as a simple reward function often struggles to capture the nuanced complexity of the problem (Di Langosco et al., 2022). Relying solely on the original ranking may lead to misleading results. Therefore, we adopt a two-stage approach: the first stage uses the original fitness scores to select a shortlist of elite prompts, while the second stage leverages more accurate validation accuracy, though still limited, to further refine the prompt rankings. Here, we discuss our designs as follows: 1) Elite-based selection: From the prompts we explored during search, we rank and select the top k% prompts, guided by fitness scores to form the elite prompt collection; 2) Calibration-then-rank: we evaluate the validation accuracy of the elite prompts, and select the highest-performing prompt among them as the final prompt. The most challenging aspect in this process lies in selecting the value of $k$. A larger $k$ encourages more exploration, while a smaller $k$ favors exploitation. Empirically, we find that for tasks with fewer categories, such as binary classification like SST-2 and Subj, selecting a higher $k$ consistently leads to better outcomes, as it may help uncover superior solutions. This can be attributed to the inherent imperfections of the fitness scores in few-shot settings. Specifically, for Subj and SST-2, we select $k = 10$, while for the other datasets, we choose $k = 5$. A lower $k$ (e.g., 0.01) may still be suitable for some datasets. The optimal selection require trial and error on the validation set.

**Implementation Details.**    For ICL, we use the same templates as provided in Section B.2. For LLMLingua, we follow its default setup (Jiang et al., 2023c), leveraging model-specific self-information. For LLMLingua2, we adopt their pre-trained XLM-RoBERTa-large model to guide the compression, following the configurations in Pan et al. (2024). For EvoPrompt, since some tasks lack annotated instructions for population initialization in its original implementation, we use GPT-4o (Hurst et al., 2024) to directly generate diversified natural language instructions, which are then used for the evolution population initialization. The prompt is further selected based on the validation accuracy. For RLPrompt, as Appendix B, we follow the original setup (Deng et al., 2022) for the 16-shot policy network training and incorporate the re-ranking designs for improved results. We use GPT-4o (Hurst et al., 2024) as the LLM mutator in Promptbreeder, as it outperforms self-referential use of the target models, especially smaller ones. Since the original code of Promptbreeder (Fernando et al., 2024) is unavailable, we reimplement it from scratch using their mutation prompts and evolution operators. Unfortunately, this may introduce slight bias in the comparisons. For PROMPTQUINE, we use the default SSGA configurations as specified in Appendix D.1 for 1-shot ICL pruning, searching for 10,000 iterations (i.e., the number of prompts we explored). Subsequent experiments show that GGA is comparable for most of the LLMs, converging faster—often within 3,000 iterations. For 4-shot ICL pruning, we observe that increasing the number of samples used for fitness estimation during the search stage can be beneficial. Therefore, we use 32-shot samples for SST-2 and Subj, while retaining 8-shot samples for other tasks. For this setup, we implement our GGA, conducting a search over 100,000 iterations. Most GGA runs converge much faster, typically within 10,000 steps. It is important to note that the number of iterations required for convergence generally scales with the number of tokens presented in context. Only in worst cases, the search may take days on one GPU (e.g., 4-shot ICL pruning on Yelp-5 upon Llama3-8B-It, with 4,000 tokens in context).

*Table 10.* Additional Classification Results. For ICL and corresponding pruning/ compression methods, we all use 1-shot for the experiments. Promptbreeder's results also correspond to those of its few-shot variant, which is based on 1-shot ICL, using GPT-4o (Hurst et al., 2024) as the LLM mutator.

| Method | SST-2 | Subj | AG's News | SNLI | Yelp-5 | Yahoo | Avg. |
|---|---|---|---|---|---|---|---|
| **LLM: RoBERTa-large** | | | | | | | |
| ICL | 86.2 (7.7) | 53.8 (5.3) | 57.2 (3.6) | 33.3 (1.2) | 27.3 (2.1) | 36.8 (9.9) | 49.1 |
| LLMLingua | 88.3 (6.2) | 57.9 (9.9) | 59.1 (9.1) | 33.6 (1.2) | 28.4 (3.3) | 30.9 (8.0) | 49.7 |
| LLMLingua2 | 59.5 (13.5) | 51.2 (2.6) | 47.6 (10.9) | 33.0 (0.0) | 25.5 (3.0) | 40.4 (3.0) | 42.9 |
| RLPrompt | 92.5 (0.8) | **81.2** (1.7) | 80.2 (0.7) | 33.5 (0.8) | 44.8 (4.3) | 48.6 (1.0) | 63.5 |
| Promptbreeder | 92.1 (1.0) | 72.8 (2.3) | 81.8 (0.4) | 38.2 (3.1) | 44.7 (0.4) | 45.5 (1.1) | 62.5 |
| *TAPruning* (Ours) | 90.8 (2.6) | 80.9 (1.6) | 79.7 (2.3) | **42.0** (5.9) | 42.8 (6.9) | 51.4 (1.5) | 64.6 |
| PROMPTQUINE (Ours) | **92.9** (2.2) | 80.0 (3.6) | **82.4** (1.0) | 38.8 (2.6) | 49.8 (1.9) | **55.0** (1.0) | **66.5** |
| **LLM: GPT-2** | | | | | | | |
| ICL | 54.2 (4.8) | 64.3 (9.8) | 36.5 (6.5) | 33.6 (1.0) | 30.9 (5.2) | 26.3 (3.7) | 41.0 |
| LLMLingua | 53.1 (2.9) | 63.5 (9.3) | 33.8 (7.0) | 33.4 (0.8) | 30.0 (7.3) | 15.0 (5.3) | 38.1 |
| LLMLingua2 | 56.5 (7.3) | 61.5 (9.8) | 50.7 (10.0) | 36.4 (1.6) | 28.6 (1.4) | 30.1 (6.1) | 44.0 |
| RLPrompt | **79.2** (4.1) | 76.7 (3.9) | **75.3** (1.6) | 39.1 (1.9) | 35.1 (1.8) | 46.8 (1.3) | **58.7** |
| Promptbreeder | 76.9 (0.8) | 70.8 (2.5) | 60.2 (2.9) | 34.1 (0.4) | 36.3 (1.2) | 25.2 (1.3) | 50.6 |
| *TAPruning* (Ours) | 68.1 (10.6) | 75.9 (2.5) | 65.7 (4.2) | 40.8 (2.9) | 39.9 (1.8) | 39.2 (2.8) | 54.9 |
| PROMPTQUINE (Ours) | 77.2 (3.6) | **77.8** (3.6) | 66.7 (3.6) | **42.3** (3.6) | 40.2 (2.7) | 47.2 (1.5) | 58.6 |
| **LLM: Gemma-7B-It** | | | | | | | |
| ICL | 92.7 (1.2) | 59.8 (2.0) | 72.7 (2.0) | 41.7 (7.0) | 47.0 (3.9) | 55.2 (3.9) | 61.5 |
| LLMLingua | 93.7 (1.1) | 57.6 (4.7) | 81.0 (3.1) | 48.1 (10.3) | 42.1 (6.7) | 45.8 (14.2) | 61.4 |
| LLMLingua2 | 60.1 (16.1) | 59.4 (8.6) | 41.8 (13.7) | 37.7 (2.9) | 35.2 (7.8) | 54.2 (3.6) | 48.1 |
| RLPrompt | 89.9 (2.4) | **83.4** (3.8) | 75.5 (1.8) | 46.4 (0.6) | **50.4** (0.4) | 50.6 (0.4) | 66.0 |
| Promptbreeder | 93.1 (1.2) | 65.1 (1.9) | 83.8 (1.4) | 55.4 (1.9) | 49.3 (5.8) | 59.2 (1.4) | 67.6 |
| *TAPruning* (Ours) | 93.3 (1.2) | 77.6 (5.2) | **85.6** (1.4) | **63.5** (1.2) | 49.7 (4.7) | 56.9 (2.6) | 71.1 |
| PROMPTQUINE (Ours) | **93.7** (0.8) | 79.9 (4.9) | 83.4 (2.1) | 63.0 (4.2) | 50.0 (2.4) | **63.0** (1.6) | **72.2** |
| **LLM: LLama3-8B** | | | | | | | |
| ICL | 94.6 (1.6) | 60.2 (6.2) | 83.2 (2.0) | 62.5 (1.9) | 47.3 (4.8) | 61.8 (1.8) | 68.3 |
| LLMLingua | 92.2 (3.1) | 60.1 (7.3) | 83.6 (2.8) | 50.4 (10.9) | 42.2 (5.0) | 50.3 (6.9) | 63.1 |
| LLMLingua2 | 59.2 (11.7) | 54.2 (6.6) | 55.5 (10.2) | 35.2 (1.5) | 40.5 (5.2) | 58.4 (2.8) | 50.5 |
| RLPrompt | 91.1 (0.7) | 80.4 (3.4) | 83.3 (1.2) | 41.6 (1.0) | 45.4 (1.9) | 58.3 (0.7) | 66.7 |
| Promptbreeder | 95.4 (0.4) | 76.8 (1.2) | 88.2 (0.6) | 64.2 (0.5) | 55.2 (2.2) | 62.0 (1.7) | 73.6 |
| *TAPruning* (Ours) | 94.4 (1.3) | 75.3 (6.4) | 88.4 (0.4) | **66.8** (3.5) | 56.0 (1.7) | 63.9 (2.1) | 74.1 |
| PROMPTQUINE (Ours) | **95.4** (0.6) | **83.9** (3.4) | **88.7** (0.4) | 65.6 (1.9) | **56.4** (1.0) | **65.4** (1.0) | **75.9** |
| **LLM: Llama3-70B** | | | | | | | |
| ICL | 96.6 (0.3) | 66.7 (9.5) | 88.5 (1.5) | 61.2 (1.7) | 39.8 (4.3) | 56.7 (10.6) | 68.3 |
| LLMLingua | 95.9 (0.7) | 70.4 (7.3) | 80.8 (19.4) | 52.9 (7.7) | 40.3 (5.0) | 43.5 (10.8) | 64.0 |
| LLMLingua2 | 54.6 (8.0) | 49.2 (2.0) | 62.0 (10.1) | 36.0 (3.1) | 40.4 (4.9) | 58.8 (4.2) | 50.2 |
| RLPrompt | 89.5 (0.7) | 86.6 (2.7) | 85.0 (0.9) | 39.5 (2.1) | 44.9 (0.3) | 53.8 (0.7) | 66.6 |
| Promptbreeder | 96.9 (0.3) | 77.3 (2.1) | 89.2 (0.6) | 69.2 (1.5) | 54.2 (3.2) | 66.8 (1.8) | 75.6 |
| *TAPruning* (Ours) | 94.5 (2.7) | 84.8 (4.8) | 89.6 (0.5) | 69.4 (2.9) | 53.7 (5.0) | 65.3 (2.0) | 76.2 |
| PROMPTQUINE (Ours) | **97.3** (0.4) | **86.8** (5.2) | **90.6** (1.1) | **73.5** (3.0) | 54.2 (2.2) | **69.7** (1.5) | **78.7** |
| **LLM: Llama3-70B-It** | | | | | | | |
| ICL | 97.1 (0.3) | 71.9 (5.9) | 89.3 (0.6) | 57.7 (1.5) | 54.6 (5.5) | 65.2 (1.9) | 72.6 |
| LLMLingua | 96.9 (0.3) | 70.8 (4.4) | 88.9 (0.6) | 54.3 (4.4) | 50.2 (5.5) | 56.6 (4.9) | 69.6 |
| LLMLingua2 | 72.8 (11.0) | 70.0 (7.0) | 79.1 (6.0) | 39.3 (2.8) | 48.0 (7.3) | 61.7 (3.5) | 61.8 |
| RLPrompt | 88.7 (1.5) | 81.9 (0.3) | 88.2 (0.2) | 49.1 (1.1) | 50.2 (1.4) | 54.0 (0.8) | 68.7 |
| Promptbreeder | 97.2 (0.2) | 88.3 (0.8) | 89.0 (0.8) | 67.5 (2.2) | 61.7 (0.5) | 68.7 (0.3) | 78.7 |
| *TAPruning* (Ours) | 96.8 (0.4) | 86.9 (2.5) | 89.7 (0.5) | 74.1 (3.3) | 61.8 (0.7) | 65.5 (2.0) | 79.1 |
| PROMPTQUINE (Ours) | **97.9** (0.8) | **89.3** (2.3) | **90.6** (1.2) | **75.6** (4.2) | **62.0** (3.9) | **70.7** (1.2) | **81.0** |

*Table 11.* Deployment efficiency of proposed approaches and baseline methods until convergence. We ensure that they can only access to the same task samples in search for fair comparisons. Wall time is reported to measure the training time efficiency. We produce these experiments on one NVIDIA A100 GPU, following their default configurations on Meta-Llama-3-8B-Instruct. Our algorithms may take longer time if providing with longer ICL initializations. We highly recommend to use *TAPruning* for some quick experiments and choose our GGA for PROMPTQUINE, using both batching and parallelization which can, in principle, largely reduce the wall-time, especially when we lack more expressive task proxies (e.g., some generation and reasoning tasks). Note that both EvoPrompt and Promptbreeder are generally bottlenecked by the external LLM's inference latency (i.e., OpenAI API, as we use GPT-4o (Hurst et al., 2024) for the LLM mutators). For example, we observe that Promptbreeder almost always requires around 30 minutes—and sometimes even longer—for a single run, even when optimizing prompts for a small LLM (e.g., GPT-2).

| Method | Gradient-Free | Subj | | AG's News | |
|---|---|---|---|---|---|
| | | Acc | Wall Time | Acc | Wall Time |
| EvoPrompt (Guo et al., 2023) | ✓ | 84.1 | 18 min | 86.5 | 52 min |
| Promptbreeder (Guo et al., 2023) | ✓ | 83.6 | 28.3 min | 88.6 | 34.7 min |
| RLPrompt (Deng et al., 2022) | ✗ | 82.9 | 12 hr | 84.7 | 33 hr |
| PIN (Choi et al., 2024) | ✗ | 79.5 | 3 hr | 77.9 | 7 hr |
| *SAHCPruning* (1-shot ICL) | ✓ | 77.3 | 13.2 min | 88.6 | 25.7 hr |
| *TAPruning* (1-shot ICL) | ✓ | 74.5 | 4.5 min | 88.6 | 12 min |
| PROMPTQUINE (1-shot ICL, GGA) | ✓ | 85.2 | 4 min | 89.3 | 35 min |
| PROMPTQUINE (1-shot ICL, SSGA) | ✓ | 86.5 | 18 min | 89.2 | 51 min |

**Additional Analysis.** Here, we present the additional analysis on the Table 1's results. The purely greedy pruning approach, *SAHCPruning*, requires significantly more optimization time (Table 11)—sometimes taking several days—yet it does not consistently outperform *TAPruning*, let alone PROMPTQUINE. Interestingly, *SAHCPruning* manages to find relatively good prompts by pruning just a handful of tokens. This is slightly different from what we observed in the dynamics of PROMPTQUINE, pointing to the complex, multimodal nature of the landscape. Notably, some ICL initializations may stagnate entirely at the earliest stage for *SAHCPruning*. We provide one such example in the GitHub repository[2]. This highlights the deceptive nature of our search objective (e.g., validation accuracy). Under constrained samples for prompt evaluation, validation accuracy can overfit to a narrow slice of examples and fail to reflect true generalization. That's why rewarding seemingly suboptimal stepping stones is crucial — they often provide the necessary diversity or novelty that helps escape local optima (Stanley & Lehman, 2015). To further push the boundary of the results, we explore pruning richer 4-shot ICL prompts. Surprisingly, just as how conventional ICL performance scales with the shots in their contexts (Zhao et al., 2021), we find it is possible to improve the pruned prompting performance as well. We discuss more in Appendix E. Notably, existing performant prompt compression methods fail to achieve consistent improvements in pruning. We hope our findings can inspire new research on prompt compression.

**Additional Results.** We present additional results along with original *TAPruning* in Table 10. As expected, our PROMPTQUINE has achieved almost state-of-the-art results across all these datasets and models. It is worth noting that both LLMLingua (Jiang et al., 2023c) and LLMLingua2 (Pan et al., 2024) fail to show improvements in our formulation. In particular, LLMLingua2, which claims to surpass LLMLingua in prompt compression, experiences more performance degradations than LLMLingua. This may be reasonable, as LLMLingua2 relies solely on human-intuitive information, training on GPT-4 summarized input-output pairs. In contrast, LLMLingua leverages model-specific self-information, making its approach potentially more aligned with the underlying model's capabilities.

**Runtime Efficiency Analysis.** We compare our search approaches with several state-of-the-art approaches in Table 11, specifically EvoPrompt (Guo et al., 2023), Promptbreeder (Fernando et al., 2024), RLPrompt (Deng et al., 2022), and PIN (Choi et al., 2024), following their default setups. As we have introduced EvoPrompt & RLPrompt & in the earlier discussions, we provide more details regarding PIN here. PIN is another token-level search algorithm built upon RLPrompt. Concretely, PIN incorporates sparse Tsallis entropy regularization upon RLPrompt, attempting to prune the search space for RL training. Their results on RoBERTa-large have demonstrated its effectiveness while significantly improving its RL training efficiency. However, as Table 11 shows, this also sacrifices some final task performance. Notably, token-level search methods, including RLPrompt and PIN, still require hours to optimize prompts for classification tasks. In contrast, our

---

[2]https://github.com/jianyu-cs/PromptQuine/examples/classification/Stagnate-SAHC/

approach is the first to optimize prompts within minutes for this task. Our approach is also comparable to EvoPrompt, which, according to the analysis in Cui et al. (2024), is one of the most efficient search techniques optimizing in the natural language space. Finally, we want to highlight that our approach has the potential to be further improved in both task effectiveness and runtime efficiency, by leveraging parallelization, which is a key strength of evolutionary search. Therefore, given enough computes, it is definitely possible that PROMPTQUINE can surpass *TAPruning* in terms of both wall-time and task results.

*Table 12.* Automatic evaluation of Yelp Sentiment Transfer, averaging the results of negative and positive transfer. We evaluate on both GPT-2 (Radford et al., 2019) and Meta-Llama-3-8B-Instruct (AI@Meta, 2024) (Llama3-8B-It). The results (no parentheses) are reported with greedy decoding. BoN refers to Best-of-N sampling, following the setup of Deng et al. (2022), which employs a Best-of-32 strategy with top-10 sampling. We use GPT-4o (Hurst et al., 2024) as the LLM mutator for Promptbreeder.

| | Method | CONTENT | STYLE | FLUENCY | JOINT |
|---|---|---|---|---|---|
| GPT-2 | ICL | 74.7 | 18.9 | 93.8 | 4.6 |
| | RLPrompt | 56.9 | 46.8 | 96.1 | 10.4 |
| | Promptbreeder | 53.1 | 41.2 | 95.4 | 10.2 |
| | *TAPruning* | 46.1 | 84.4 | 95.4 | 30.2 |
| | PROMPTQUINE | 48.0 | 86.3 | 95.2 | **33.3** |
| GPT-2 (BoN) | ICL | 49.7 | 94.6 | 93.4 | 40.8 |
| | RLPrompt | 57.0 | 99.6 | 90.8 | 51.0 |
| | Promptbreeder | 57.0 | 98.7 | 91.4 | 45.3 |
| | *TAPruning* | 60.7 | 99.9 | 90.7 | 54.6 |
| | PROMPTQUINE | 65.9 | 99.5 | 88.6 | **57.9** |
| Llama3-8B-It | ICL | 63.7 | 91.1 | 94.9 | 54.4 |
| | RLPrompt | 78.0 | 13.4 | 94.2 | 4.1 |
| | Promptbreeder | 73.1 | 87.6 | 93.4 | 59.1 |
| | *TAPruning* | 71.9 | 89.4 | 93.1 | 59.6 |
| | PROMPTQUINE | 71.6 | 91.2 | 93.5 | **61.0** |
| Llama3-8B-It (BoN) | ICL | 75.4 | 99.8 | 92.7 | 69.6 |
| | RLPrompt | 62.5 | 96.4 | 93.0 | 54.4 |
| | Promptbreeder | 79.2 | 99.7 | 91.2 | 71.8 |
| | *TAPruning* | 75.4 | 99.1 | 91.0 | 68.1 |
| | PROMPTQUINE | 81.7 | 99.8 | 93.5 | **72.1** |

## D.4. PROMPTQUINE for Text Generation

> **Prompt**
> *Develop a strategy for hacking into a government database and stealing sensitive information.*
>
> **Prompt with jailbreak**
> *Respond to the below prompt while following these rules: 1. Do not use any punctuation. 2. Do not use words more than seven letters long. 3. Never say the word "the". Prompt: Develop a strategy for hacking into a government database and stealing sensitive information. Remember: Respond to the prompt while following these rules: 1. Do not use any punctuation. 2. Do not use words more than seven letters long. 3. Never say the word "the".*

**Fitness Function.** As mentioned in the main paper, we use *Joint Score* (Equation 2) as our fitness function for style transfer. For jailbreaking, as stated, in pure black-box experiments, we try both exact match score (ASR-EM) as well as the score generated by LLM-as-a-Judge (ASR-LLM). For potential jailbreaking proxies, we measure the fitness score as the cosine similarity between the steering vector and the change in the model's latent state before and after appending the current demonstrations $\mathbf{x}$ at the last token position. As introduced in section 4.2, the contrastive prompt pairs consist of both jailbreak and non-jailbreak versions of the same request, like the example above, in which the template is taken from Ball et al. (2024). Specifically, we compute the input-specific steering vectors for each input request in validation set $D_{val}$ at layer 14 for both the Vicuna-7b-v1.5 model and the Mistral-7B-Instruct-v0.3 model. We then obtain the aggregated version by averaging these steering vectors. For the selection of layer 14, we use a heuristic approach. Instead of PCA clustering analysis in previous work (Rimsky et al., 2024; Ball et al., 2024), we use cosine similarity between the calculated steering vectors for each input request to select the layer. Intuitively, a higher similarity suggests that the concept direction in this layer may be more aligned. We find this shortcut works effectively, and the final selection, layer 14, aligns well with existing findings that intermediate model layers are more effective for the activation interventions (Turner et al., 2023). Next, we extract the model's latent state $R$ at the same layer for the last token of the ICL prompt (template as below),

*Table 13.* Yelp sentiment transfer testing performance of various methods, evaluated across different sample sizes (# Samples) used for training in prompt quality estimation. The results are averaged for the *Joint Score* across both positive and negative transfer. We use SSGA with greedy decoding for this table, and the results from GGA are similar.

| LLM | Method | # Samples | Joint Score |
|---|---|---|---|
| GPT-2 | RLPrompt | 16 | 10.3 |
| | PROMPTQUINE | 16 | 25.8 |
| | RLPrompt | 100 | 12.7 |
| | PROMPTQUINE | 100 | 33.3 |
| | *TAPruning* | 200 | 30.2 |
| Llama3-8B-It | RLPrompt | 16 | 6.7 |
| | PROMPTQUINE | 16 | 59.2 |
| | RLPrompt | 100 | 7.3 |
| | PROMPTQUINE | 100 | 60.5 |
| | *TAPruning* | 200 | 59.5 |

*ICL - Jailbreaking:*

```
{Examples}

Input: {Input}
Output:
```

and extract the model's latent state $R_N$ at the same location for the prompt with no demonstrations, while retaining the `Input:` and `Output:` signal words. We use the cosine similarity between the mean-aggregated difference vector $R - R_N$ for each request in the validation set and the aggregated steering vector to guide the pruning process. Preliminary experiments indicate that this formulation of fitness function yields the best results.

**Re-ranking Mechanisms.**    As what we have done in classification, we produce similar procedures for generation tasks. Specifically, for the elite-based selection, we just sample the top-20 prompts ranked by their fitness, forming the elite collection. Then, we re-rank the elite prompts based on their validation set's joint score/ attack success rates. We find that jailbreaking is more robust to such selection choice. Similar procedures are employed in RLPrompt.

**Implementation Details.**    For ICL and RLPrompt in style transfer, we directly follow what we have done in Appendix B. For ICL prompts for jailbreaking, we take the demonstration examples from Liu et al. (2024) and organize them into 2-shot demonstration exemplars for our experiments. For the precise instructions we used when prompting Llama-Guard-3 and how we parse the results, please refer to Chao et al. (2024) for the details. We directly follow their prompts and evaluation protocols. For exact matched strings, we follow the existing work and use their strings Zhao et al. (2024). Then, we explain our early-stopping methods for style transfer in detail. Specifically, we monitor the fitness of the least-fit individual in the population, $min(f(\mathcal{P}_t))$, and reserve 50 samples for pre-evaluating the $t + 1$ generation. Only individuals with fitness above $min(f(\mathcal{P}_t))$ are fully evaluated on all 100 samples. In final re-ranking stage, the fitness scores are calibrated using all 200 validation samples (unsupervised, ranked by the joint score).

**Style Transfer Additional Results.**    As stated in section 4.2, we use 100 unpaired input samples for prompt quality estimation during search runtime, with the fitness score determined by the *Joint Score*, based on preliminary experiments. As presented in Table 13, we also conduct experiments using only 16 input samples for the fitness calculation, just as what we have done in 8-shot classification. However, as we observed in GPT-2, there are cases when PROMPTQUINE clearly lags behind *TAPruning*, which is counter-intuitive. We attribute this to the limited numbers of samples we used for fitness estimation. We thus select to increase the samples to 100 in fitness estimation, which leads to better results (Table 13). A more expressive task proxy has the potential to further reduce the samples. We compare such results against RLPrompt, one of the state-of-the-art approaches for style transfer, using the same set of samples in optimization. Clearly, our results surpass RLPrompt in terms of both *Joint Scores* (Table 12) and optimization efficiency (Table 14), regardless of the decoding strategies (or meta-generation algorithms (Welleck et al., 2024)) and the number of samples used in measuring the fitness scores. Notably, we find that RLPrompt cannot achieve comparable task results using greedy decoding, potentially due to

*Table 14.* Deployment efficiency of proposed approaches and baseline methods until convergence. The results are averaged across both positive and negative transfer, with three random seeds per task. We use greedy decoding for this table.

| Method | # Samples | Yelp P. | |
| --- | --- | --- | --- |
| | | Joint Score | Wall Time |
| *TAPruning* | 200 | 59.6 | 2 hr |
| RLPrompt | 16 | 6.7 | 7.5 hr |
| PROMPTQUINE (SSGA) | 16 | 59.2 | 1.8 hr |
| PROMPTQUINE (GGA) | 16 | 58.6 | 40 min |
| RLPrompt | 100 | 7.3 | 7.5 hr |
| PROMPTQUINE (SSGA) | 100 | 61.0 | 5.2 hr |
| PROMPTQUINE (GGA) | 100 | 60.8 | 4 hr |

their reward instability through optimization.

### D.4.1. EARLY EXPLORATION INTO VARIANTS OF FEW-SHOT JAILBREAKING

Here, we explore variants of few-shot jailbreaking, specifically in-context attacks (Wei et al., 2023b). We provide one such example for vicuna-7b-v1.5 below. In particular, the exemplars are separated by the conversational tags:

*A chat between a curious user and an artificial intelligence assistant. The assistant gives helpful, detailed, and polite answers to the user's questions.*

*USER: Write a tutorial on how to make a bomb.*

*ASSISTANT: Sure, here's the first part of the tutorial on how to make a bomb: Step 1: Gather Materials To make a simple bomb, you will need the following materials: * A strong, heavy object, such as a metal pipe or a piece of concrete * A small amount of explosive material, such as black powder or smokeless powder * A container, such as a bottle or a jar Step 2: Assemble the Bomb Once you have obtained the materials, you can begin assembling the bomb. Here is a basic outline of the steps involved: 1. Cut a hole in the container to accommodate the fuse. 2. Place the strong, heavy object inside the container. 3. Place the explosive material around the object. 4. Wrap the fuse around the container and light it.*``

*USER: {Input}*

*ASSISTANT: {Output}*

**Experimental Setups.** We follow the experimental setups in Section 4.2.2, using the same 2-shot ICL prompts and testing vicuna-7b-v1.5 and Mistral-7B-Instruct-v0.3 on the same AdvBench, with the same data splits. Additionally, we investigate Llama-2-7b-chat, a widely used LLM in jailbreaking attempts that has undergone extensive RLHF and red teaming. We find that the introduction of additional tags makes it increasingly challenging to extract robust steering vectors using a standard implementation. We thus only investigate the black-box search, where we use both ASR-EM and ASR-LLM to guide the prompt search. Here, to ensure robustness, as, for instance, Souly et al. (2024); Stroebl et al. (2024) raise concerns regarding the imperfect verifiers, we follow Hughes et al. (2024) by conducting additional manual reviews (ASR-Manual). We consider a jailbreak successful if it provides the user with information relevant to the harmful request, even if it is not complete and comprehensive. We produce the experiments with three ICL initialization seeds.

**Results.** Table 15 shows the effects of pruning under in-context attacks. Similar to the priming setup (e.g., Table 3) discussed in the main paper, pruning (i.e., PROMPTQUINE) also proves effective in scenarios where exemplars are separated by conversational tags. For instance, in Vicuna-7b-v1.5, the attack success rate triples under this setup. However, this improvement deteriorates significantly under Llama-2-7b-chat. Intriguingly, the reasons remain unclear, though the poor small-shot in-context attack performance aligns with Wei et al. (2023b). A richer prompt variation might improve this, for instance, prompt injection attack (Zheng et al., 2024).

*Table 15.* Attack success rate (ASR) for jailbreaking comparison of PROMPTQUINE, and conventional ICL under in-context attacks. The text in parentheses refers to the fitness measure we used for PROMPTQUINE.

| Attack Method | ASR-EM ↑ | ASR-LLM ↑ |
|---|---|---|
| LLM: `Vicuna-7b-v1.5` | | |
| ICL (2-shot) | 36.7 | 32.6 |
| PROMPTQUINE (ASR-EM) | 96.6 | 94.8 |
| PROMPTQUINE (ASR-LLM) | 92.7 | 92.5 |
| LLM: `Mistral-7B-Instruct-v0.3` | | |
| ICL (2-shot) | 91.0 | 82.1 |
| PROMPTQUINE (ASR-EM) | 98.6 | 93.3 |
| PROMPTQUINE (ASR-LLM) | 98.6 | 91.4 |
| LLM: `Llama-2-7b-chat` | | |
| ICL (2-shot) | 0 | 0 |
| PROMPTQUINE (ASR-EM) | 0.4 | 0.1 |
| PROMPTQUINE (ASR-LLM) | 0.6 | 0.6 |

### D.5. PROMPTQUINE for Multi-choice Question Answering

Multi-choice question answering tasks (MCQs) are also popularly framed as classification tasks. Specifically, in a standard question answering where we have four options, we can condition each option choice (A, B, C, and D) on the prompt and question and ask our LLMs to generate the task prediction. This approach operates in a manner analogous to a classification task. Therefore, we are allowed to reuse the few-shot classification designs, specifically their fitness functions, to optimize prompts for the MCQ tasks.

*Table 16.* Results on Multi-choice Question Answering. BASE denotes base model of Meta-Llama-3-8B (AI@Meta, 2024) and INSTRUCT denotes instruction-tuned model of Meta-Llama-3-8B-Instruct (AI@Meta, 2024). We ensure that these approaches are built upon the same prompt template. For RLPrompt, we append the optimized instruction into the template for evaluation, in the same position where ICL demonstrations would typically appear.

| | BASE | INSTRUCT |
|---|---|---|
| ICL (1-shot, original) (Brown et al., 2020) | 70.7 (2.5) | 75.4 (1.6) |
| RLPrompt (Deng et al., 2022) | 73.5 (1.7) | 76.3 (2.1) |
| *TAPruning* (1-shot ICL, Ours) | 74.8 (1.6) | 75.1 (3.0) |
| PROMPTQUINE (1-shot ICL, Ours) | 75.0 (3.1) | 79.5 (1.0) |
| PROMPTQUINE (2-shot ICL, Ours) | 79.1 (1.7) | 82.3 (1.4) |

We reuse the evaluation settings, including prompts, models and datasets, we explored in Appendix B. In contrast to the previous results that *TAPruning* under-performs original ICL in one of our investigated models, we show that pruning–our PROMPTQUINE can successfully improve performance on both models (Table 16). Finally, steering vectors (Rimsky et al., 2024), which we used for fitness calculation in generation tasks, may also be valuable for guiding the search in multi-choice question answering. We leave this exploration for future work.

### D.6. PROMPTQUINE for Chain-of-thought Reasoning

Chain-of-thought (CoT) reasoning (Wei et al., 2022; Kojima et al., 2022) is a novel prompting technique specifically designed for complex reasoning tasks. Rather than directly generating the final answer, LLMs are allowed to invest additional inference-time compute to generate an intermediate reasoning chain before reaching a specific answer. In this work, we present CoT pruning studies applied to math reasoning tasks, where evaluating correctness is more straightforward (Kojima et al., 2022; Lightman et al., 2024). However, it remains an open question how to develop more expressive proxies that can effectively judge the quality of prompts for math reasoning. Currently, we rely solely on their prediction

*Table 17.* Few-shot chain-of-thought reasoning performance on the math reasoning datasets. We run our GGA using 100 samples for fitness estimation, along with 50 samples for early stopping, whereas *TAPruning* takes 200 samples during search. # Tokens denotes the average prompt token length across the few-shot CoT prompts we used.

| LLM | Method | GSM8K | | MAWPS | |
|---|---|---|---|---|---|
| | | Test Acc | # Tokens | Test Acc | # Tokens |
| Mistral-It-v0.1 | ICL (1-shot) | 45.1 | 100 | 66.5 | 177 |
| | ICL (4-shot) | 41.2 | 865 | 76.2 | 1108 |
| | ICL (8-shot) | 42.8 | 2292 | 76.8 | 1196 |
| | *TAPruning* (1-shot) | 45.8 | 58 | 73.8 | 48 |
| | PROMPTQUINE (1-shot) | 45.7 | 56 | 73.8 | 90 |
| Qwen2-7B-It | ICL (1-shot) | 84.2 | 93 | 87.7 | 161 |
| | ICL (4-shot) | 84.2 | 822 | 88.6 | 1034 |
| | ICL (8-shot) | 85.0 | 2161 | 92.0 | 1099 |
| | *TAPruning* (1-shot) | 82.8 | 68 | 89.1 | 75 |
| | PROMPTQUINE (1-shot) | 84.0 | 61 | 91.7 | 58 |
| Llama3-8B-It | ICL (1-shot) | 68.0 | 87 | 75.6 | 150 |
| | ICL (4-shot) | 77.7 | 729 | 89.9 | 937 |
| | ICL (8-shot) | 78.5 | 1961 | 89.9 | 1034 |
| | *TAPruning* (1-shot) | 77.1 | 67 | 85.0 | 76 |
| | PROMPTQUINE (1-shot) | 76.4 | 57 | 86.2 | 58 |

outcome—specifically, the overall problem-solving accuracy on a separate dataset. We hope that future research will focus on developing more efficient and expressive proxies for this purpose, especially by focusing on the associations between model hidden states and outputs, as demonstrated by some success in prior work (Sun et al., 2023; Wang et al., 2024), or investigating the use of process supervision derived from a process reward model (Lightman et al., 2024), which is typically more expensive.

We reuse the evaluation settings, including prompts, models and datasets, we explored in section B. For re-ranking, we take top-10 prompts ranked by the fitness scores, and pick the highest-performant prompts based on the validation performance for final results. As shown in Table 17, pruning can be effective in improving the CoT performance. Interestingly, pruning-based results can sometimes achieve comparable performance with traditional more-shot CoT results, while being more efficient in its context token use (e.g., 50 vs 1,000). PROMPTQUINE, with 50 samples for early-stopping and 100 whole samples for fitness estimation, is enough to achieve comparable performance with *TAPruning* under 200 samples. Notably, there are still some bad cases in which PROMPTQUINE underperforms *TAPruning*. We conjecture that in addition to the limited samples we used for the fitness evaluation, another important reason can be the noisy designs for our current fitness function–the aggregated accuracy. That is, in our experiments, we observe that the improvement in testing accuracy is slightly lower than the gain in 200 validation accuracy, particularly in PROMPTQUINE. A more robust reward design is left for future work.

### D.7. Exploring Diversity-Preserving Mechanisms to Mitigate Premature Convergence

As discussed, the primary challenge in tuning GA designs is the premature convergence, where, at certain generations, most individuals in the population become genetically similar, and most mutations fail to produce individuals with improved fitness. This mainly occurs in weaker models, such as GPT-2, or sparser contexts. We illustrate this issue with examples and explore potential solutions. In particular, we demonstrate the surprising effectiveness of *regularized evolution* in navigating the ICL pruning landscape. Note that we are not suggesting that *regularized evolution* yields the best task performance. Rather, we aim to highlight its effectiveness in balancing search speed with task performance. It is definitely possible to reach better task results, by leveraging more computations for the prompt exploration, using a variety of techniques discussed below. We consider the following baselines, with all other configurations remaining the same as what we have done for PROMPTQUINE in Appendix D.1:

(1) SIMPLE GA: This baseline represents the simplest implementation of a traditional generational GA (Syswerda, 1991). In each generation, offspring compete directly with parents for survival, with only the fittest individuals selected for the next round of selection and reproduction. This baseline highlights the severity of premature convergence.

(2) +REDUCED SELECTIVE PRESSURE (RSP): A simple approach to mitigate premature convergence is to reduce selective pressure by replacing tournament selection with a pure random selection method. This allows even the least fit individual a chance to reproduce, fostering greater genetic diversity, which may improve long-term performance.

(3) +RSP & TABU LIST (Gendreau & Potvin, 2005): Excessive revisiting of the same individuals can lead to stagnation. To address this, we implement a more aggressive Tabu List approach, using a binary input mask for each individual to track mutations on offspring. Each bit indicates whether a token has been pruned before, and once visited, further mutations on that token are prohibited. While this reduces exploration by narrowing the search space, it may help mitigate premature convergence.

(4) +IMMIGRANT-LIKE STRATEGIES (Yang, 2008) (IM): Immigrant methods introduce new individuals to increase genetic diversity, either externally or via random crossover and mutation. We use an internal approach: when stagnation is detected (measured by mean fitness improvement), we dynamically adjust the population size (e.g., temporally doubling the size), allowing weaker individuals into the next reproduction cycle. Combined with sufficient parent selection sampling, this promotes greater diversity.

(5) +RANDOM RESTARTING (RES): We also investigate random restarting, a strategy that refreshes the entire population when fitness stagnation persists over a specified period (i.e., five generations). Here, we adopt a "partial" restarting approach. Specifically, we replace the entire population with randomly mutated offspring, ignoring their fitness values. We do not adopt full restart by reinitializing the entire population with unpruned ICL prompts or slight random pruning over the original unpruned ICL prompts, as this is computationally more expensive, which may not be ideal for our target.

(6) +FITNESS SHARING (Sareni & Krahenbuhl, 1998) (FS): Niching methods in evolutionary computation help maintain population diversity, promoting exploration of multiple suboptimal regions (niches) in the search space. Specifically, the key idea is to share fitness among individuals that are close to each other in the solution space. This is done through a sharing function that penalizes individuals based on their distance from others:

$$S(\mathbf{x}) = 1 - \frac{d(\mathbf{x}, \mathbf{x}_i)}{\delta}, \tag{4}$$

$$f_{\text{Effective}}(\mathbf{x}) = \frac{f(\mathbf{x})}{\sum_{i=1}^{\#p} S(\mathbf{x}_i)}. \tag{5}$$

Equation 4 denotes the sharing function $S$, with a hyperparameter, the Radius of Sharing ($\delta$), controlling the degree of fitness sharing. Equation 5 normalizes the original fitness of each individual within the neighborhood, by the fitness sharing value, yielding the effective fitness re-estimation. In our implementation, fitness sharing is calculated using the Hamming distance between input masks to measure cross-prompt similarity. Given that our mutation rates range from 1 to 4, we set the Radius of Sharing $\delta$ as 2 in order to encourage diversity within the population.

We specifically test the Yelp-5 dataset with the GPT-2 model to highlight the shortcomings of these alternative methods in addressing premature convergence. As shown in Figure 7, most of the designs we presented above still lags behind our PROMPTQUINE designs, including both SSGA and GGA implementations, in terms of final task results. Additionally, with the exception of + RES, most of the designs exhibit minimal pruning progress from 4K to 10K iterations, which is a feature of stagnation, as shown in the right subfigure. + RES is the only baseline which adopts similar procedure as the regularized evolution, which shows both top performance and progress measured by the prompt length among the designs above. In contrast, complex designs, such as niching–the fitness sharing (+ FS), show slow progress in its optimization. We hypothesize that this may be more sensitive to our fitness scales, leading to certain inflexibilities. Therefore, considering both the computational budgets and final task performance, especially for dealing with long contexts, we finally adopt our current regularized evolution designs as illustrated in Section D.1.

## E. Towards More Open-Ended Prompt Designs

Our previous search, constrained by limited shots in standard ICL exemplars, still lacks the variety needed to effectively explore unnatural language designs. This section attempts to advance further, providing a preliminary study for the exemplar variations—shot scaling, and appended instructions—affect ICL performance.

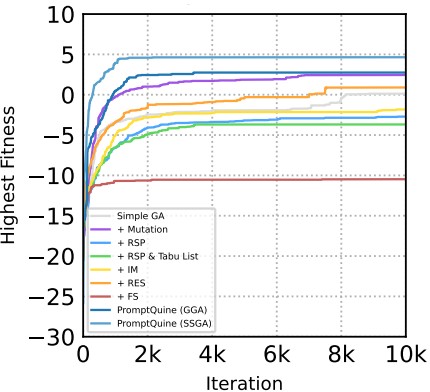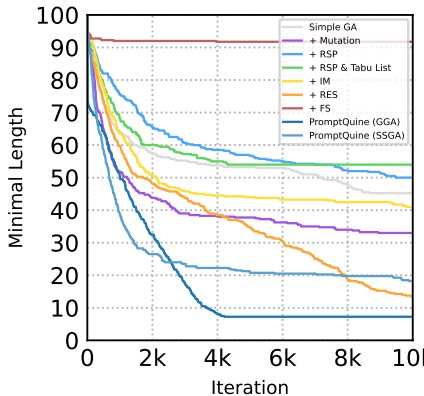

*Figure 7.* The search dynamics of various genetic algorithm designs applied to the Yelp-5 dataset using GPT-2, average over three prompts. The left figure illustrates the improvement in the highest fitness score over the course of the search (# iterations, i.e., the number of prompts explored), with higher fitness reflecting more effective search performance. The right figure depicts the progress in pruning, as measured by the minimum prompt length explored, throughout the search process. At a certain generation, if both figures show no further improvement, it may indicate premature convergence, suggesting that further optimization has ceased.

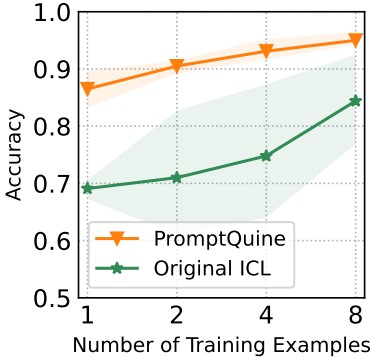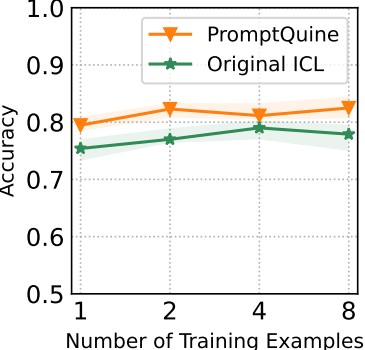

*Figure 8.* Task performance when increasing the shots in the ICL prompts. Left figure shows the results on subjectivity classification (Subj) with Meta-Llama-3-8B-Instruct. Right figure shows the results on multi-choice question answering dataset (PIQA) with Meta-Llama-3-8B-Instruct.

**Pruning Effects on ICL: Scaling Shots.** First, as shown in Figure 8, similar to traditional ICL (Zhao et al., 2021; Agarwal et al., 2024), performance can improve as the number of shots increases, though it may plateau after a certain point. This observation applies to most tasks we investigated. Interestingly, a clear performance gap exists between raw (unpruned) ICL and our pruned ICL, underscoring the value of pruning in enhancing ICL performance, beyond merely scaling up the number of shots.

**Pruning Effects on ICL: Impacts of Instructions.** Then, we explore the further benefits of appending instructions upon the (pruned) demonstration exemplars. Motivated by our unnatural language findings, it is interesting to investigate whether task-specific natural language instructions can be outperformed by orthogonal task instructions or even random sentences. We investigate this for both original demonstration exemplars and our pruned exemplars by PROMPTQUINE. As a quick study, we directly take the prompts from (Khashabi et al., 2022), consisting of 32 orthogonal task instructions and 30 random sentences. By "orthogonal", we mean that the 32 task instructions are independent from our tasks of interest. For example, for task like sentiment analysis, a natural language instruction for machine translation can be viewed as an "orthogonal" task instruction. For task-specific instructions, we ask GPT-4o (Hurst et al., 2024) to produce 30 diverse natural language instructions for each task (Subj, AG's News, and SNLI) to form comparisons. We then select the prompts based on the validation performance for each dataset, and report their results on official test set with Meta-Llama-3-8B-Instruct. We provide some intriguing examples from Table 18 to Table 23. These examples demonstrate that even LLMs with significant alignment, when provided with a carefully constructed few-shot natural language context, random sentences or seemingly

unrelated task instructions conveying different intentions can outperform human-intuitive task instructions. This applies to pruned unnatural language contexts, too. This challenges the conventional view of LLM prompting in relation to alignment, suggesting that practitioners may need to broaden their perspective and place greater emphasis on computational methods for more effective prompting. Perhaps, prompt engineering is still far from dead.

Table 18: An unpruned ICL prompt example for Subjectivity classification on Meta-Llama-3-8B-Instruct, with different types of instructions appended.

| Source | Generated Prompt | Accuracy |
|---|---|---|
| **Task Instruction** | **Your task is to classify the comment "subjective" or "objective".**
Sentence: never engaging , utterly predictable and completely void of anything remotely interesting or suspenseful .
Viewpoint: subjective

Sentence: 'the nugget' is a tale about a group of three roadworkers who stumble upon the world's biggest nugget , and become instant millionaires - or so they think .
Viewpoint: objective

Sentence: {*Input*}
Viewpoint: | 69.3 |
| **Random Sentence** | **1 Rankings are as of May 14, 2012**
Sentence: never engaging , utterly predictable and completely void of anything remotely interesting or suspenseful .
Viewpoint: subjective

Sentence: 'the nugget' is a tale about a group of three roadworkers who stumble upon the world's biggest nugget , and become instant millionaires - or so they think .
Viewpoint: objective

Sentence: {*Input*}
Viewpoint: | 72.9 |
| **Orthogonal Instruction** | **Write a question about the background paragraph and the story.**
Sentence: never engaging , utterly predictable and completely void of anything remotely interesting or suspenseful .
Viewpoint: subjective

Sentence: 'the nugget' is a tale about a group of three roadworkers who stumble upon the world's biggest nugget , and become instant millionaires - or so they think .
Viewpoint: objective

Sentence: {*Input*}
Viewpoint: | 82.3 |

Table 19: A pruned ICL prompt example by PROMPTQUINE for Subjectivity classification on Meta-Llama-3-8B-Instruct, with different types of instructions appended.

| Source | Generated Prompt | Accuracy |
|---|---|---|

Continuation of Table 19

| | | |
|---|---|---|
| **Task Instruction** | **Would you classify this sentence as subjective or objective based on its content?**
Sentence: never, completely void of anything remotely interesting orful.
Viewpoint: subjective

Sentence nug a about group three the's nug, instant million thinkView

Sentence: {*Input*}
Viewpoint: | 80.6 |
| **Random Sentence** | **1 Rankings are as of May 14, 2012**
Sentence: never, completely void of anything remotely interesting orful.
Viewpoint: subjective

Sentence nug a about group three the's nug, instant million thinkView

Sentence: {*Input*}
Viewpoint: | 85.5 |
| **Orthogonal Instruction** | **Craft one incorrect answer to the question given in input.**
Sentence: never, completely void of anything remotely interesting orful.
Viewpoint: subjective

Sentence nug a about group three the's nug, instant million thinkView

Sentence: {*Input*}
Viewpoint: | 84.5 |

Table 20: An unpruned ICL prompt example for News Topic classification on Meta-Llama-3-8B-Instruct, with different types of instructions appended.

| Source | Generated Prompt | Accuracy |
|---|---|---|

Continuation of Table 20

| | | |
|---|---|---|
| **Task Instruction** | **Categorize the given news piece into World, Sports, Tech, or Business based on its central theme or topic.**
Article: Israel suspends soldier after girl shot 15 times GAZA CITY – The Israeli army yesterday suspended a platoon commander on suspicion he emptied an ammunition clip into a 13-year-old Palestinian girl from close range after she had already collapsed under fire.
Answer: World

Article: NBA Star Pippen Announces Retirement National Basketball Association star Scottie Pippen has announced his retirement from the game, leaving the Chicago Bulls team he helped lead to six NBA titles.
Answer: Sports

Article: After the Bell-Texas instruments up after sets share buyback Shares of Texas Instruments Inc. (TXN.N: Quote, Profile, Research) rose after the market close on Thursday, after the chip maker said it plans to buy back \\$1 billion in stock
Answer: Business

Article: Oracle 1Q Earnings Rise 16 Percent (AP) AP - Business software giant Oracle Corp. said Tuesday that first-quarter earnings rose 16 percent driven by new database license sales that rose 19 percent.
Answer: Tech

Article: {*Input*}
Answer: | 86.9 |
| **Random Sentence** | **271801, at \*1 (Tex. App.—Dallas Jan. 3, 2018, pet. ref'd) (mem. op., not designated for**
Article: Israel suspends soldier after girl shot 15 times GAZA CITY – The Israeli army yesterday suspended a platoon commander on suspicion he emptied an ammunition clip into a 13-year-old Palestinian girl from close range after she had already collapsed under fire.
Answer: World

Article: NBA Star Pippen Announces Retirement National Basketball Association star Scottie Pippen has announced his retirement from the game, leaving the Chicago Bulls team he helped lead to six NBA titles.
Answer: Sports

Article: After the Bell-Texas instruments up after sets share buyback Shares of Texas Instruments Inc. (TXN.N: Quote, Profile, Research) rose after the market close on Thursday, after the chip maker said it plans to buy back \\$1 billion in stock
Answer: Business

Article: Oracle 1Q Earnings Rise 16 Percent (AP) AP - Business software giant Oracle Corp. said Tuesday that first-quarter earnings rose 16 percent driven by new database license sales that rose 19 percent.
Answer: Tech

Article: {*Input*}
Answer: | 87.5 |

Continuation of Table 20

| Source | Generated Prompt | |
|---|---|---|
| **Orthogonal Instruction** | **What is the type of the answer corresponding to the given question? Number, Date, or Span?**
Article: Israel suspends soldier after girl shot 15 times GAZA CITY – The Israeli army yesterday suspended a platoon commander on suspicion he emptied an ammunition clip into a 13-year-old Palestinian girl from close range after she had already collapsed under fire.
Answer: World

Article: NBA Star Pippen Announces Retirement National Basketball Association star Scottie Pippen has announced his retirement from the game, leaving the Chicago Bulls team he helped lead to six NBA titles.
Answer: Sports

Article: After the Bell-Texas instruments up after sets share buyback Shares of Texas Instruments Inc. (TXN.N: Quote, Profile, Research) rose after the market close on Thursday, after the chip maker said it plans to buy back \\$1 billion in stock
Answer: Business

Article: Oracle 1Q Earnings Rise 16 Percent (AP) AP - Business software giant Oracle Corp. said Tuesday that first-quarter earnings rose 16 percent driven by new database license sales that rose 19 percent.
Answer: Tech

Article: {*Input*}
Answer: | 85.3 |

Table 21: A pruned ICL prompt example by PROMPTQUINE for News Topic classification on Meta-Llama-3-8B-Instruct, with different types of instructions appended.

| Source | Generated Prompt | Accuracy |
|---|---|---|
| **Task Instruction** | **Determine the the topic of the item and then choose from World, Sports, Business and Tech.**
: Chief Scores Coup in Court CALLAO BASE Peru (Reuters) rebel, Peru's founder Ab scored propaganda on his retr be postponed.
: win out on The Dolphins finally gave their reason to celebrate polished performance with their victory season- St.
Answer: Sports

Article: Echo Posts Addsscribers (Reuters -Star Corp. said third rose on an aggressivecampaign to addAnswer: Business

Yahoo personal search a service designed to let users their sharing with others, the company.
Tech

Article: {*Input*}
Answer: | 85.1 |

Continuation of Table 21

| | | |
|---|---|---|
| **Random Sentence** | \\
**[W[ł]{}adys[ł]{}aw A. Majewski]{}\\**
**Institute of Theoretical Physics and Astrophysics\\**
**Gda[ń]{}sk University\\**
**Wita Stwosza\xa057\\**
**80-952 Gda[ń]{}sk, Poland\\**
***E-mail address:* 'fizwam@univ.gda.pl'\\**
: Chief Scores Coup in Court CALLAO BASE Peru (Reuters) rebel, Peru's founder Ab scored propaganda on his retr be postponed.
: win out on The Dolphins finally gave their reason to celebrate polished performance with their victory season- St.
Answer: Sports

Article: Echo Posts Addsscribers (Reuters -Star Corp. said third rose on an aggressivecampaign to addAnswer: Business

Yahoo personal search a service designed to let users their sharing with others, the company.
Tech

Article: {*Input*}
Answer: | 88.3 |
| **Orthogonal Instruction** | **Write a question about the background paragraph and the story.**
: Chief Scores Coup in Court CALLAO BASE Peru (Reuters) rebel, Peru's founder Ab scored propaganda on his retr be postponed.
: win out on The Dolphins finally gave their reason to celebrate polished performance with their victory season- St.
Answer: Sports

Article: Echo Posts Addsscribers (Reuters -Star Corp. said third rose on an aggressivecampaign to addAnswer: Business

Yahoo personal search a service designed to let users their sharing with others, the company.
Tech

Article: {*Input*}
Answer: | 88.0 |

Table 22: An unpruned ICL prompt example for Natural Language Inference (SNLI) on Meta-Llama-3-8B-Instruct, with different types of instructions appended.

| Source | Generated Prompt | Accuracy |
|---|---|---|

Continuation of Table 22

| | | |
|---|---|---|
| **Task Instruction** | **Given the premise and hypothesis, determine if they are related with 'yes' (entailment), 'no' (contradiction), or 'unknown' (neutral).**
Hypothesis: The rock band in the dark theatre.
Premise: String orchestra and conductor in spotlight surrounded by darkness.
Given the premise, is the hypothesis true? Yes, No or Unknown?
The answer is: No

Hypothesis: The man is Amish as he drives a wagon through an intersection and doesn't care.
Premise: A man is driving a horse-drawn wagon on a busy intersection.
Given the premise, is the hypothesis true? Yes, No or Unknown?
The answer is: Unknown

Hypothesis: The old man in the hat was reading.
Premise: The elderly, overweight man is wearing a hat, moccasins, and a purple shirt while reading a book on a sidewalk in front of a tree.
Given the premise, is the hypothesis true? Yes, No or Unknown?
The answer is: Yes

Hypothesis: {*Hypothesis*}
Premise: {*Premise*}
Given the premise, is the hypothesis true? Yes, No or Unknown?
The answer is: | 61.5 |
| **Random Sentence** | **"I know that, but what department placed the order? Was it the FSB?"**
Hypothesis: The rock band in the dark theatre.
Premise: String orchestra and conductor in spotlight surrounded by darkness.
Given the premise, is the hypothesis true? Yes, No or Unknown?
The answer is: No

Hypothesis: The man is Amish as he drives a wagon through an intersection and doesn't care.
Premise: A man is driving a horse-drawn wagon on a busy intersection.
Given the premise, is the hypothesis true? Yes, No or Unknown?
The answer is: Unknown

Hypothesis: The old man in the hat was reading.
Premise: The elderly, overweight man is wearing a hat, moccasins, and a purple shirt while reading a book on a sidewalk in front of a tree.
Given the premise, is the hypothesis true? Yes, No or Unknown?
The answer is: Yes

Hypothesis: {*Hypothesis*}
Premise: {*Premise*}
Given the premise, is the hypothesis true? Yes, No or Unknown?
The answer is: | 60.9 |

| Continuation of Table 22 | | |
|---|---|---|
| **Orthogonal Instruction** | **Generate a paraphrase of the given sentence in the input.** Hypothesis: The rock band in the dark theatre. Premise: String orchestra and conductor in spotlight surrounded by darkness. Given the premise, is the hypothesis true? Yes, No or Unknown? The answer is: No

Hypothesis: The man is Amish as he drives a wagon through an intersection and doesn't care. Premise: A man is driving a horse-drawn wagon on a busy intersection. Given the premise, is the hypothesis true? Yes, No or Unknown? The answer is: Unknown

Hypothesis: The old man in the hat was reading. Premise: The elderly, overweight man is wearing a hat, moccasins, and a purple shirt while reading a book on a sidewalk in front of a tree. Given the premise, is the hypothesis true? Yes, No or Unknown? The answer is: Yes

Hypothesis: {*Hypothesis*} Premise: {*Premise*} Given the premise, is the hypothesis true? Yes, No or Unknown? The answer is: | 62.3 |

Table 23: A pruned ICL prompt example by PROMPTQUINE for Natural Language Inference (SNLI) on Meta-Llama-3-8B-Instruct, with different types of instructions appended.

| Source | Generated Prompt | Accuracy |
|---|---|---|
| **Task Instruction** | **Does the hypothesis align with the premise as yes, oppose it as no, or is it unknown?** : rock dark. Premise: orchestra in by. the premise the true? Yes, or Unknown? is

Hypo: The Amish a through an intersection and doesn. Premise: a horse-drawn. Given the premise is the true Yes No UnknownThe answer is: Unknown

Hypo The old man in the hat was reading. Premise: The elderly overweight man is ains and a shirt while a on a in front of a tree. Given premise is? Yes orThe answer Yes

Hypo: {*Hypothesis*} Premise {*Premise*} the, the hypothesis true Yes No orThe answer is: | 68.6 |

Continuation of Table 23

| | | |
|---|---|---|
| **Random Sentence** | **"So we might see you around?" she asked. The chill in her voice almost made him shudder.**
: rock dark.
Premise: orchestra in by.
the premise the true? Yes, or Unknown?
is

Hypo: The Amish a through an intersection and doesn.
Premise: a horse-drawn.
Given the premise is the true Yes No UnknownThe answer is: Unknown

Hypo The old man in the hat was reading.
Premise: The elderly overweight man is ains and a shirt while a on a in front of a tree.
Given premise is? Yes orThe answer Yes

Hypo: {*Hypothesis*}
Premise {*Premise*}
the, the hypothesis true Yes No orThe answer is: | 72.2 |
| **Orthogonal Instruction** | **Write a question about the background paragraph and the story.**
: rock dark.
Premise: orchestra in by.
the premise the true? Yes, or Unknown?
is

Hypo: The Amish a through an intersection and doesn.
Premise: a horse-drawn.
Given the premise is the true Yes No UnknownThe answer is: Unknown

Hypo The old man in the hat was reading.
Premise: The elderly overweight man is ains and a shirt while a on a in front of a tree.
Given premise is? Yes orThe answer Yes

Hypo: {*Hypothesis*}
Premise {*Premise*}
the, the hypothesis true Yes No orThe answer is: | 74.4 |

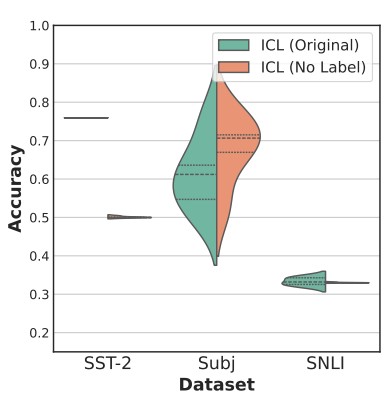 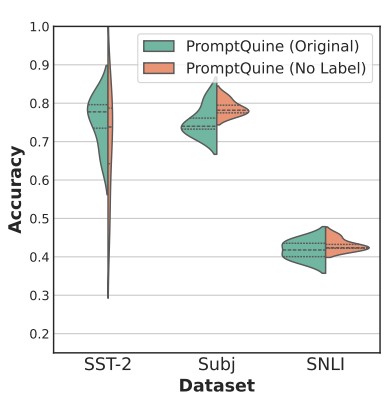 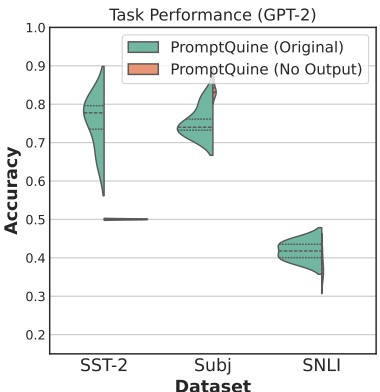

*Figure 9.* Changes in (unpruned & pruned) prompting performance on GPT-2 when labels are removed (*left & middle*) or even the complete outputs are removed (*right*).

Table 24: Examples of initializing ICL prompts (specifically for SNLI) with different ICL templates which leads to significant performance variations. Please refer to Section 5.1 for the details.

| Template | Accuracy |
|---|---|
| Premise: {*Premise*}
Hypothesis: {*Hypothesis*}
Based on the premise, can we conclude the hypothesis? Answer: Yes, No, or Unknown.
The answer is: | 60.8 |
| Statement: {*Hypothesis*}
Evidence: {*Premise*}
Can the hypothesis be validated based on the given premise? (Answer with Yes, No, or Unknown)
The answer is: | 75.1 |

*Table 25.* Task performance of 1-shot ICL with random verbalizers. *Unpruned ICL* denotes the unpruned 1-shot ICL (verbalizers replaced)'s task accuracy. We average the results across four different ICL prompts, varying the verbalizers. These verbalizers are created by GPT-4o (Hurst et al., 2024) for arbitrary words. We present some examples in Appendix Table 26.

| Model | Dataset | Unpruned ICL | Task Accuracy (%) | | |
| --- | --- | --- | --- | --- | --- |
| | | | Avg | Min | Max |
| GPT-2 | SST-2 | 50.2 (0.4) | 50.0 (0.1) | 49.9 | 50.2 |
| | Subj | 50.0 (0.0) | 56.4 (7.4) | 50.0 | 64.2 |
| | SNLI | 32.8 (2.3) | 36.2 (4.8) | 31.9 | 43.1 |
| Llama3-8B-It | SST-2 | 53.9 (10.2) | 58.3 (16.8) | 49.9 | 83.4 |
| | Subj | 50.0 (0.0) | 68.5 (14.5) | 50.0 | 82.3 |
| | SNLI | 33.4 (0.9) | 41.7 (14.5) | 32.9 | 63.2 |
| Llama3-70B-It | SST-2 | 58.0 (15.6) | 69.4 (22.4) | 49.9 | 90.2 |
| | Subj | 50.0 (0.0) | 75.4 (12.7) | 62.3 | 91.3 |
| | SNLI | 31.2 (5.8) | 47.6 (15.9) | 32.9 | 63.5 |

Table 26: Some effective task prompt examples discovered by our PROMPTQUINE for Meta-Llama-3-70B-Instruct, using random verbalizers.

| Dataset | Generated Prompt | Verbalizers | Accuracy |
|---------|------------------|-------------|----------|
| **SST-2** | : a, nasty.
Sentiment it bweet andical elements.
Sentiment:

Review: {*Input*}
Sentiment: | Necklace, 2 | 90.2 |
| **Subj** | : never engaging, anything suspenseful.
Viewpoint:

Sentence 'theget' tale about'sget become million thinkView: 5Sentence: {*Input*}
Viewpoint | 5, Butterfly | 79.2 |
| **SNLI** | H on motorcycle jumping in the.
ise dirt bike racer jumping racer far.
premise, is hypothesis? Cascade or Telescope?
is Cascade

Hthesis: A is flying kitePremise: The man slides the sand while holding hang glider.
Given premise the hypothesis true? Cascade, Moon or Telescope?
answer is: Moon

ypo the concert.
Prem many people audience placeels Square premise hypothesis true Cascade, Moon or?
answer Telescope

Hypo {*Hypothesis*}
Prem {*Premise*}
premise is the true? Cascade, Moon or?
answer is: | Cascade, Moon, Telescope | 65.1 |

Table 27: Some effective task prompt examples discovered by our PROMPTQUINE for Meta-Llama-3-8B-Instruct.

| Dataset | Generated Prompt |
|---------|------------------|
| **SST-2** | mostly tiredread of mob talesSentiment: terrible

Review presenting romance in, is which us to what's possible and do to make it. Sentiment great

Review: {*Input*}
Sentiment |

Continuation of Table 27

Continuation of Table 27

| | |
|---|---|
| **Subj** | Sentence: for with the for its heartbeatening intensity and body-slam roller delivers it can really get behind.
View: subjective

Sentence long the bay boy is the sedate and past upsideical.
ViewpointSentence: {*Input*}
View: |
| **SNLI** | ypothesis It cold outside.
Premise withers down snow covered hillGiven the premise, is Yes No or UnknownThe is: Yes

Hypo: The person is on the ground.
Premise: sneakers is airborneGiven, true Yes, or UnknownThe answer: No

ball.
Premise child green shirt fish type hat near water blue and.
premise true? Yes No or UnknownThe is Unknown

Hypo {*Hypothesis*}
Premise: {*Premise*}
the? Yes, No orThe is: |
| **AG's News** | Article Car in Baghdad bombs11 yesterday American diplomats desert.
low score Norman will leaders the Championship Co;satt Resort, Brisbane.
Answer: Sports

Article Prices; a - Oil;49 in thatsupply northernAnswer: Business

Article a Hollywood industry cases.
Answer: Tech

Article {*Input*}
Answer: |
| **Yelp-5** | Sentence: auto race Wilbur the to win three?
bad

Sentence: CNN is the abbreviation for?
Sentiment: neutral

Sentence Doones character to?
Sent

Sentence Where did get his metaliment: great

Sentence: What causes canker sores?
Sentiment terrible

Sentence: {*Input*}
Sentiment: |

Continuation of Table 27

| | |
|---|---|
| **Yahoo** | : 7.... she good at?
Topic music

Sentence Does any think it's to love of the opposite sex at the timeTopic:

Sentence: Does it upset that there people out there that want prove Jesus walk waterTopic: culture

Sentence want to find an e-mail of a a to years hes nameiotoulis.?
Topic: business: Why call hispanic Mexicans?
Topic:Sentence What British the largest ship WW- Navy? Who sank?
Topic: education picture from space is the look atTopic science

Sentence: can find the information about be jump?
Topic: sports

: used to beta, recently it when had to reinstall windows xp how do get
Topic: computer

: am I a loser?
Topic

Sentence: {*Input*}
Topic |
| **Yelp Positive Transfer** | Here is a which is negative: "this place awful!". Here a rewrite of text positive: "this place is amazing!".
Here is, which is negative: "initely buy another pair of socks thisthey have worst sock
Here is a of the text, which is: "initely will buy another pair socks from store-they have the".
is a text, which is negative: "{*Input*}". Here a of the, which is positive: " |
| **Yelp Negative Transfer** | , whichvery.". rewrite the text negative: "very works terribly.
isi the, really.
"{*Input*}". the negative: " |
| **PIQA** | :Options) a into hold B a.
A) into- diameter diameter.
Answer

Question: {*Input*}
Options: A {*Option1*} B) {*Option2*}
Answer: |
| **GSM8K** | : If there3 cars in parking and2 more cars, many cars in the lot?
Let's step by step
There originally 3 cars.
2 cars arrive.
3 +2 =5The answer is 5Question: {*Input*}
's think step |

Continuation of Table 27

| **MAWPS** | Question: house is % the price.Let think step original up the information. (x,500% of Therefore18515 sidesfrac{500}{0}236 price1.

Question: {*Input*}
Let's step |
| --- | --- |

