# OpenReview forum: "Evolving Prompts In-Context: An Open-ended, Self-replicating Perspective"
_ICML.cc/2025/Conference — ICML 2025 poster_

### Official Review · Reviewer_e5jB · 2025-02-21

**Overall Recommendation:** 4

**Summary:**

The paper proposes PromptQuine, an automated prompt optimization strategy that prunes a given prompt using evolutionary search to improve the performance at a given task. The method outperforms existing methods when validated on classification, multi-choice question answering and reasoning datasets across a wide range of models. Moreover, it is more efficient than previously proposed methods.

**Claims And Evidence:**

PromptQuine claims to outperform all other prompt optimization methods. The considered models and datasets are extensive enough. However, some important baselines like gradient-based methods (Autoprompt or other variants like GCG) are missing. Moreover, the method is said to be efficient but it takes much more time to run compared to the greedy pruning baseline which has similar performance.

**Essential References Not Discussed:**

None

**Experimental Designs Or Analyses:**

The baselines are not well-tuned (see questions).
The 1-shot ICL setting is misleading. You are still using multiple examples for fitness estimation which effectively extract information form these examples. If multiple examples are used for PromptQuine, Best-of-N selection should be used for manual prompts, TAPruning or greedy pruning for fair comparison.
The analysis section is no very substantial. Apart from the task label and signal words, there is no quantitative analysis about the kind of tokens that are pruned by the algorithm. This might help us better understand what kind of tokens models are more sensitive to.

**Methods And Evaluation Criteria:**

The method is evaluated on a wide range of datasets using models of varying sizes (from 350M to 70B parameters) and architectures (encoders and decoders).

**Other Comments Or Suggestions:**

None

**Other Strengths And Weaknesses:**

Strengths:
- The paper contains a detailed appendix explaining in great detail all the experiments.

Weaknesses:
- TAPruning seems to be a weaker version of greedy pruning which is not constrained by the order of the tokens to be pruned. The significance of the proposed method is rather limited as its performance is still very close to the baseline while being far less efficient (as show in Table 10).
 Concerning clarity, some parts of the paper like section 5.2 are rather hard to read and seem detached from the rest of the paper.

**Questions For Authors:**

1- What is the 4-shot performance of TAPruning? It is not reported in Table 2.

2- What is the performance of greedily pruning the tokens? (no left-to-right pruning order as in TAPruning but select the best position to prune each time) Please also include the 4-shot performance of this method.

3 - What is the performance of the baselines when using the same number of examples as PromptQuine? For example, best-of-n selection could be used with manual prompting and greedy pruning.

4 - How does a gradient-based method (like Autoprompt or other varians like GCG) compare to PromptQuine?

5 - Why don't you include the latest approaches (like gradient-based ones) for jailbreaking LLMs in your experiments? What is the main advantage of PromptQuine compared to these methods? PromptQuine relies on steering vectors which require access to the language model.

**Relation To Broader Scientific Literature:**

The paper proposes a prompt search technique. It is a prompt engineering method that seeks to maximize performance without modifying the model weights. It is also related to feature attribution and how LMs respond to unnatural language.

**Theoretical Claims:**

None

---

> ### Author Rebuttal · Authors · 2025-03-28
>
> We greatly appreciate Reviewer e5jB for suggestions on our experimental setups. We are delighted that you acknowledge the **richness in detail** of our study. We address your concerns below:
>
> Anonymous link (AL) for several tables: https://anonymous.4open.science/r/ughj/e5jB/README.md
> >Recap: Claims & Objective
>
> We argue we didn't claim *Quine outperforms all others*, in abstract, we use "Gibberish always matches or surpasses," in introduction, we use "Pruning can near SOTA," and in each section, we present results objectively. **Instead**, our goal is to deliver **insight that simple ICL pruning** can be on par with work which use various external information, e.g., +tokens, countering common view and contributing to AI emergence from constraints [1,2]. *Quine/TAP are just what we strive to improve under pruning (pursue knowledge over SOTA)*. Also, comparing search algorithms solely by numbers is limiting, as scalable methods e.g. ES can further improve on the same prune landscape. We'll thus add a figure for sample efficiency, comparing TA/ES/greedy.
> >Models from 350M
>
> We started from GPT2, 125M (Table 10).
> >Not tuned baselines?
>
> Pruning results in Table 2 are reported w/ varied ICL initializations, where ICL baseline are direct comparisons. We can't perform BoN across seeds.
> * BoN for TAPruning/"Greedy": We **stated in Algo 2, which used held-out valid for BoN (same samples)**.
> * BoN for ICL: The only setup available is to construct exemplars using samples from held-out valid (e.g.,200). It costs many computes(N=51^4*4!=16568064 for 1-shot agnews, taking **4 yrs for a 1-shot dataset BoN w/ vLLM**). We are unable to support so. That is also why all existing work by validation BoN [3,4+] don't do such enumeration and report ICL as us. We follow such convention. Your request requires other principled algos, beyond our work. We show Bo10 under limited samples in AL.
>
> >gradient baselines?(Gb)
>
> Good questions! Let's clarify.
> * White-box (wb) jailbreaking: Using steering vector (SV) does **NOT** imply wb is required. Sorry for ambiguity. We indeed view it hard for Pruning (line362-sparse, line365-potentials), thus we explore w/ a small-scale priming study for dense signals w/ tools from interpretability(SV). We'll enhance paper: show black-box/bbx results first and explore SV. We provide such results in AL README(+GCG result). Besides, we plan to explore further under varied ICL setups in paper (see reply to 8j6A), for rich picture of pruning.
>
> * Other tasks: bbx is enough, and Gb is empirically weak under FSL[6,7].
>
> >Not substantial Analysis
>
> We agree the insight is intriguing for analysis. Section 6 presents label words in ICL, where [8,9+] exclusively address it in full papers. Pruned token analysis remains hard, with only viable hypothesis (we) being whether function words are pruned more—an insight we deemed limited, as noted in abstract and intro, let alone by human intuition. Yet, we do plan to add studies for ICL stabilization, inspired by KfU1. Given current density, we leave others for future work.
> >Code Files
>
> Apologies for the tight schedule. We clearly state this in README, which we'll release+refactor soon or upon request.
> >TAPruning weaker "greedy pruning"?
>
> *We respectfully disagree*. Greedy search may refer to steepest-ascent hill-climbing (SAHC), which has limitations: 1.Search landscape can be multimodal and deceptive (section 4.2), where SAHC can stagnate and TA escapes with speed up [10]. 2.Rigorously, TA has solution regions where SAHC won't touch.
> **Case where greedy stagnates entirely**: See AL traces.csv. SAHC is weaker than TA for no progress on this prompt, limiting its generality. We provide new greedy/TA results in AL, where SAHC takes *days* for search.
> >ES no significance?
>
> *We respectfully disagree*, ES is indeed a pioneer study for handling multimodal landscape. We stated in line250, agreed by other reviewers.
> * Efficiency: TAPruning is indeed our algo. In comparison, Quine is on par with published work in terms of runtime efficiency (Table10). Also it's the first token-level search that can optimize in mins for some tasks(line366). While TAPruning dominate runtime on one GPU, ES has potentials to be parallelized [11] (e.g., reproduction), largely reducing runtime (e.g., line90). We'll make what we claimed clear for "better runtime" in Table10.
> * Performance: We argue there're consistent improvements over TA across models in nearly all tables, up to 12%, large when converted to #corrects. Please also refer to TAPruning 4-shot in AL README, where TA/Greedy(e.g., 11% lower) are all weaker than ES. We'll put greedy in paper to enhance why ES for broad open-endedness/deception. [10].
>
> [1] Collective Intelligence for DL [2] Open-Endedness is Essential for ASI [3] AutoPrompt [4] GCG [5] RLPrompt [6] TEMPERA [7] ICL Learns Label Relationships yet not.. [8] Larger LMs Do ICL Differently [9] Abandon Objectives [10] Why Greatness cannot be planned [11] ES as a Scalable Alternative to RL

---

> > ### Comment · Reviewer_e5jB · 2025-04-02
> >
> > All the questions were answered by the authors. They also addressed all the concerns raised in the review. I will update the score accordingly. The community would benefit from the contributions in the paper but its clarity ought to be extensively improved. Some important aspects of the algorithms such as efficiency are not emphasized well enough in the paper. Some terms are misleading and underspecified.

---

> > > ### Author Response · Authors · 2025-04-03
> > >
> > > Many thanks for the updated score, and thanks for your agreements on the potential impacts of this work's findings towards overall community. As we have responded to almost all reviewers, we will **definitely improve the paper presentation and expand some studies** inspired by all reviewers for **a significantly enhanced paper**. Thanks for the invaluable time on our work!

---

### Official Review · Reviewer_8j6A · 2025-02-24

**Overall Recommendation:** 4

**Summary:**

The paper introduces a novel prompt design paradigm that challenges the conventional approach of using well-crafted natural language prompts for large language models. The authors demonstrate that pruning random demonstrations into seemingly incoherent "gibberish" can significantly improve performance across a variety of tasks, including classification, multi-choice question answering, generation, and reasoning.

This effect is shown to generalize across different LLMs, regardless of their alignment. The authors propose a self-discover prompt optimization framework called PromptQuine, which employs evolutionary search to automatically identify effective pruning strategies.

The paper provides extensive empirical evidence supporting these findings and discusses their implications for understanding in-context learning and prompt design in LLMs.

## update after rebuttal

I thank the authors for the rebuttal. I think this is an interesting paper, but I am keeping my score because the limited mechanistic analysis (Section 6) leaves the "why" of pruning’s success underexplored.

**Claims And Evidence:**

The claims in the paper are generally well-supported by the empirical results presented in both the main text and the appendix. The authors assert that pruning random demonstrations into "gibberish" enhances LLM performance across diverse tasks, and they provide results in tables (e.g., Table 1, Table 2) showing improvements over baselines like RLPrompt across multiple datasets and models.

However, the claim that pruned prompts work better lacks a comprehensive mechanistic explanation. While section 6 offers some analysis on the role of label words, the evidence is limited and does not fully elucidate why pruning enhances performance.

**Essential References Not Discussed:**

The paper cites relevant prior work adequately.

**Experimental Designs Or Analyses:**

The experimental designs appear sound and robust. The authors evaluate their method across many tasks and LLMs, using datasets detailed in Appendix C (Table 6) and comparing against baselines like RLPrompt, LLMLingua, and EvoPrompt.

**Methods And Evaluation Criteria:**

The proposed method, PromptQuine, utilizes an evolutionary search framework based on genetic algorithms to optimize prompt pruning, which is a sensible approach for the discrete optimization problem of finding effective prompt subsequences. This choice aligns with the problem's combinatorial nature, where gradient-based methods are less applicable.

Evaluation criteria include standard metrics such as accuracy for classification and math reasoning.

**Other Comments Or Suggestions:**

Typos: In the abstract, "let alone human intuitions." should likely be "let alone human intuition"

**Other Strengths And Weaknesses:**

### Strengths

- Originality: The idea of pruning demonstrations into "gibberish" seems novel, creatively challenging the norm of natural language prompts and offering a fresh perspective on ICL optimization.
- Empirical Rigor: Extensive experiments across tasks and models provide strong evidence of the method’s effectiveness and generalizability.
- Significance: The findings could inspire new directions in prompt design and ICL stabilization, as noted in the conclusions.

### Weaknesses

- Clarity: The paper is dense, with many details relegated to the appendix.
- Analysis Depth: The limited mechanistic analysis (Section 6) leaves the "why" of pruning’s success underexplored.

**Questions For Authors:**

Can you provide more details on the computational resources required for PromptQuine (e.g., GPU hours)?

**Relation To Broader Scientific Literature:**

The paper situates its contributions within the literature on prompt optimization and in-context learning (ICL).

**Theoretical Claims:**

The paper does not present theoretical claims or formal proofs. It is primarily an empirical study focused on demonstrating the efficacy of the pruning-based approach.

---

> ### Author Rebuttal · Authors · 2025-03-30
>
> We are *very grateful* to Reviewer 8j6A for their highly positive comments on our paper. Thanks for acknowledging our efforts on making this paper *empirically rigor* and the **Significance, Creativity and Originality** of this work towards general *prompt tuning and ICL*. We will **make every effort to further improve** the paper, especially the presentations. We address your questions below:
> >Analysis depth
>
> Thanks for the interest.
> * **Organize potential research questions into Future work/ Implication**: We also share this with nearly all reviewers. As our work is dense, we plan to leave many of the questions as future work. Indeed, various interpretability questions can arise from our findings, and many of them remain challenging open problems, and we believe each deserves a separate research paper, especially why pruning is effective and why such ICL behavior could work. We hold strong belief addressing them can largely advance the field.
> * **Incorporate additional analysis studies**: Inspired by KfU1, we'll add studies showing the failures of ICL pruning, e.g.. while effective for ICL stabilization, the template selection [1] still matters. Further, as some interests may center around jailbreaking, we'll expand studies on that, e.g., extending studies on varied ICL setups, e.g., in-context attack [2], prompt injection attack [3] which go beyond current priming setup. We plan to analyze this challenging task, where the most common metric provides a sparse reward signal and many other formulation exists (e.g., tools for mechanistic interpretability--steering vectors and output joint probability as gradient methods).
>
> >Computational Resources
>
> After further check, we find that we leave many descriptions in text, instead of making explicitly clear in Table 10/12, e.g., LLM with Llama3-it, GPU type, etc. We'll refine the captions. To give a brief overview, for search, in classification/QA tasks, on a single A100 GPU, TAPruning/PromptQuine can run within an hour for low-shot ICL. As parallelization can be an option [4] for PromptQuine, runtime efficiency can improve with more computes. In generation, including reasoning, they take hours as shown in Table 12. TAPruning, which based on first-choice hill climbing, can always be a good start point for particular tasks of interest, as they converge quicker in one GPU. Dealing with long contexts typically take more time, e.g., hours or even days, yet may increase final result. That could be one tradeoff. *One potential follow-up work is to improve the performance/efficiency for long contexts*. Some intriguing starting points could be attempting to exploit model-specific information, e.g., it is also interesting that LLMLingua outperforms LLMLingua2 in this compression as guided search formulation. In summary, we now claim this framework can achieve *decent runtime efficiency*, as what we replied to KfU1.
> >Writing Issues
>
> Sorry for the typos. We also find some places that can be improved, including the one suggested by you and KRwQ. We'll ensure that all these will be enhanced/ fixed!
> >Clarity, too dense
>
> Apologize for this. As what we replied to KfU1 and KRwQ, we'll **follow all your suggestions** to further structure and highlight the takeaways, and make all settings clear to readers.
>
> [1] Quantifying LMs’ Sensitivity to Spurious Features in Prompt Design or: How I learned to start worrying about prompt formatting, ICLR [2] Jailbreak and Guard Aligned LMs with Only Few In-Context Demonstrations, arxiv [3] Improved Few-Shot Jailbreaking Can Circumvent Aligned Language Models and Their Defenses, NeurIPS [4] Evolution Strategies as a Scalable Alternative to RL, arxiv

---

### Official Review · Reviewer_KRwQ · 2025-03-13

**Overall Recommendation:** 3

**Summary:**

The paper applies evolutionary algorithms to the paradigm of LLM prompt pruning, introducing a new algorithm called PromptQuine. PromptQuine autonomously searches for better pruning strategies through an iterative process of mutation and selection, inspired by biological self-replication and evolutionary dynamics. It demonstrates effectiveness across diverse tasks, consistently outperforming existing prompt optimization methods.

**Claims And Evidence:**

Most claims are supported with clear and convincing evidence.

Figure 1 (right), “ES exhibit greater robustness to task difficulty, such as increasing the number of shots, which amplifies solution sparsity.” (grammatical mistake here, should be ES exhibits) and line 237, “the success ratio of RS approaches zero as task difficulty increases”. In Figure 1 (right), I only see one line plotted, and am unsure whether that line is ES or RS. How are the claims derived from Figure 1 (right)?

The paper claims that PromptQuine automatically optimizes the pruning strategies. However, the mutation that pruning strategies undergo is very limited, i.e., having mutations in the pruning tokens by randomly flipping the bits. This means that the search space may not fully explore more complex pruning strategies. As a result, while PromptQuine demonstrates strong empirical performance, its optimization approach might be constrained to local improvements rather than a truly open-ended exploration of prompt space.

There is also the claim that incoherent “gibberish” or “secret language” can be more effective prompts than well-crafted ones. However, the usage of these terms is vague and imprecise, lacking a clear definition or formal characterization. Furthermore, since the mutation strategy and search space in PromptQuine is so limited, i.e., word order is still kept, “gibberish” and “secret language” seems like an exaggeration of the prompts produced.

**Essential References Not Discussed:**

Since the key contribution is the application of evolutionary algorithms to prompt design, some recent related works are missing from the citations. Specifically, “Promptbreeder: Self-Referential Self-Improvement via Prompt Evolution” and “Evolution through Large Models”. Promptbreeder could be a good approach to compare PromptQuine against too. Furthermore, since Promptbreeder works in a larger search space than PromptQuine, I think this work should definitely be discussed in the paper.

**Experimental Designs Or Analyses:**

All experiments are sound with repeated runs and variance.

**Methods And Evaluation Criteria:**

Yes. The authors compare PromptQuine against competitive baselines across a diverse set of tasks.

**Other Comments Or Suggestions:**

For Figure 3, it is difficult to see the differences in performance between the models. Truncating the starting percentage of the plots (e.g., displaying only values from 75% onwards) would improve clarity and make the differences more noticeable.

In line 202, “we follow (Krishna et al, 2020; …)”. The citations should not be in parentheses.

Similar to the usage of “gibberish” and “secret language”, the term “conventional wisdom” is also used vaguely. While Wan et al. (2024) is cited as a reference, it is unclear to what extent this truly represents a conventional understanding scientifically. A more precise explanation or broader citation of prior work would help establish its validity.

**Other Strengths And Weaknesses:**

Strengths:
- I think this paper is a significant contribution and shows interesting results, that just by pruning prompts, PromptQuine is able to achieve better results across a diverse set of benchmarks.

Weaknesses:
- See above. The overall writing is good, but some of the claims and terminologies used are very vague and ambiguous.

**Questions For Authors:**

1. In the abstract, line 31, what does it mean by “low-shot regimes”?
2. Optimization challenges in sparse, multimodal landscapes is not a new challenge in evolutionary algorithms. Are there related works that show the same challenges?
3. To mitigate evaluation noise, re-ranking mechanisms are used. There are related evolutionary works that deal with noise and stochasticity, e.g., Fast and stable MAP-Elites in noisy domains using deep grids.
4. While the paper briefly touches on its limitations and potential future work throughout, it would be more useful to include a dedicated section in the conclusion to discuss these aspects in greater detail.

**Relation To Broader Scientific Literature:**

The paper builds on prior work in prompt optimization, evolutionary search, and in-context learning. The paper proposes PromptQuine, an evolutionary algorithm for prompt pruning. It extends research on automatic prompt optimization (e.g., RLPrompt, LLMLingua) by using mutation and selection rather than RL or token attribution. The study also connects to evolutionary search in NLP and aligns with findings on LLM sensitivity to prompt design.

**Theoretical Claims:**

The definition and usage of “Partial Context Hypothesis” are vague. Terms such as “potentially redundant contexts” and “well-specified natural language prompts” are not carefully defined, making it unclear what specific aspects of a prompt contribute to redundancy or well-specification. Despite this lack of clarity, the hypothesis is referenced multiple times throughout the experimental analysis. A more precise definition and formal criteria for evaluating context redundancy and prompt specificity would strengthen the validity of the hypothesis.

---

> ### Author Rebuttal · Authors · 2025-03-28
>
> We want to express our sincere thanks to Reviewer KRwQ for their detailed tips to improve our writing. **This is invaluable, and we do learn a lot.** We are also grateful for viewing our work as **significant contribution** by presenting *interesting results* to the community. Here is the Table/Fig link: https://anonymous.4open.science/r/ughj/KRwQ/README.md
> >Vague Terminologies
>
> * **SecretLang & Gibberish**: Apologize for not explicitly defining "secret language" and now clarify it in a new paragraph (Section 2.2) following its intro in Para. 1.*Specifically, secret language refers to unnatural language whose syntax and semantics are incoherent and difficult for humans to parse, yet can be surprisingly effective in certain scenarios. In the absence of ..., such prompts are typically regarded as mysterious, hidden, and inherently non-scalable.* Further, the secret language we use in paper refers to prompts generated by prior methods, e.g., RLPrompt, which limits by algorithmic capacity. For the use of gibberish texts, we discuss in intro (line 30) as *syntactically and semantically strange*, which are used less. Hope this won't cause overclaims.
> * **Partial Context Hypothesis**: We agree. So we plan to use formal mathematical languages which avoid use of "redundant" and "well-specified". Intuition: Given a natural prompt (e.g., ICL) x={...} with performance X and a unnatural prompt z (by existing SOTA, e.g. RLPrompt) with Z, is it possible to prune a few tokens {...} which leads to prompt y with enhanced performance Y, which potentially outperforms Z. y can be purely syntactically and semantically unnatural language.
> * **Conventional Wisdom**: will cite OpenAI guide and [1,2+] for ICL stabilization (i.e., well-tuned). As also inspired by KfU1, we'll add a ICL study highlighting some structured insights. Sorry for the pointer, due to word limits.
>
> >PromptBreeder+
>
> We cited PB but not explicitly discussed. We'll discuss in related work for any work related to Open-endedness & LLMs. Specifically, we'll explain how ours differs from PB. Re comparisons, we thought before. Unfortunately, they didn't release code for use. As ES involves varied designs, we are unsure if reproduction fully reflects PB's result. In replying to KfU1, we reproduce with numerical results. We may report in paper. For larger space, we generally agree, but want to emphasize: While PB introduces token variations, the search space in our pruning context is also large, typically O(n!). Considering the discrete nature, even a slight token drop can cause large changes, which can be different from ES in continuous space. So I feel that both space by current discrete algo is large, and both **framework** is scalable yet constrained by computations.
> >Search Space
>
> Regarding pruning strategies, we admit that bit-flip can be limited in its representation power. We will introduce an ablation study (by KfU1) for mutation rates, e.g., why up to 4 for **solution sparsity**. That is also why we retain word orders, as we observed that many lead to meaningless mutations under selection pressures (+a study). Regarding open-ended exploration of prompt space, we agree and argue for not fully aligned w/ our target (rethink prompt space). The most relevant term is the *open-ended search* (established OES) we highlight in abstract. That is an insight we aim to deliver, e.g., intriguing yet unnatural prompts remain hidden, lurking just within our reach. We suggest the field to pay more attention to OES (cf. [3,4,5,6]), jumping out of linguistics for diverse, novel **unnatural language**.
> >Optimization challenges
>
> We argue that Section 4.2 is meant to motivate ES, and the landscape analysis is also new as the insight of pruning towards such large improvement is also new. Also, to answer e5jB, we'll discuss the deception of held-out objective [6,7], which **necessitate reward suboptimal stepping stones** like ES in paper.
> >Other Writing Questions
>
> Fig1 is inspired by [8] (Fig4), and we will add relative success rate (RS over/divided by ES) descriptions in caption and polish terminology in section 4.2. Regarding typos/LaTex/future work, we will follow the tips, correct all and state clearly. Regarding low-shot regimes, we borrow this in few-shot learning [9,10], in sense that we search using only low-shot samples, e.g., fitness measure. We can avoid such jargon. Further, we apologize for the ambiguity around noise; We use it to refer to deceptive/imperfect reward (not uncertainty/noise in DG MAP-Elites, thanks!), i.e., stronger prompts are rated as lower ranks, which we address using held-out score (rerank), thought still being limited. We'll make it clear. Fig3:https://anonymous.4open.science/r/ughj/KRwQ/re_Fig3.pdf
>
> [1] Fantastically Ordered Prompts [2] Learning To Retrieve for ICL [3] Minimal criterion coevolution [4] Randomly Wired NN [5] Weight Agnostic NN [6] Abandon Objectives [7] Go-Explore [8] AutoML-Zero [9] Pi-Tuning [10] Loss landscape is all your need

---

### Official Review · Reviewer_KfU1 · 2025-03-17

**Overall Recommendation:** 4

**Summary:**

This paper introduces an evolutionary method called PromptQuine for optimizing few-shot prompts by pruning them. They show that their optimized prompts outperform the original few-shot prompts as well as the RLPrompt baseline on a held-out set of test examples across a wide range of standard language model benchmarks. Interestingly, the resulting pruned prompts are often gibberish text (similar to what was found in other prior work) but retain some key features, including reference to the task at hand.

**Claims And Evidence:**

The evaluation setup appears to be correct, although it could be better explained. In particular, it is imperative that the examples on which the prompts are pruned (training / validation splits) are distinct from the examples on which they are evaluated (test split). This is alluded to in Evaluation Settings (line 182) but it is not sufficiently clearly stated. It would be very useful if the authors can separate out an Evaluation section which pertains to the whole paper and which clearly states the nature of the held-out evaluation, include a specific step-by-step example, so that the reader can ensure the rigour of the evaluation procedure.

The pilot study with hill-climbing search is well-explained and the claims are substantiated by the evidence in this section.

The results for PromptQuine itself are more questionable. The authors choose to only present the positive results in the main text, with the negative results confined to the Appendix (D5 and D6) without further discussion in the main text. In my opinion the authors should be more transparent about the fact that PromptQuine underperforms RLPrompt in various tasks in the main text, and provide some qualitative insights / analyses as to why that might be case.

Similarly I find Section 5.2 a little more difficult to justify as "convincing evidence". The authors introduce a new metric with which to measure performance in jailbreaking, and it is hard to be certain that this metric has not been chosen so as to demonstrate the strength of their method. I would recommend that they use existing metrics / benchmarks in the literature to the extent possible.

The "task label" analysis appears convincing and gives good insight into which parts of the pruned prompts are important. However it is not sufficiently well explained. How are "task label" words identified? Also in Figure 4, the variance appears quite high and this deserves commentary. How much can we conclude from such high-variance experiments?

The examples in Table 16 are useful, and it would be good to see more of these analysed in the main text, for both situations where PromptQuine works and situations where it does not.

**Essential References Not Discussed:**

See response to previous question.

**Experimental Designs Or Analyses:**

I have already commented on this above.

**Methods And Evaluation Criteria:**

The choice of benchmark datasets are sensible, although it would be interesting to see results on some more modern tasks too e.g. GPQA and MMLU.

The methods used are well-motivated. However the explanation of the evolutionary algorithm is long-winded, and too much is made of what is essentially a very standard method. It would suffice to describe the algorithm as a standard evolutionary algorithm (citing a textbook in the literature) and then specify the mutation and selection operators, along with the definition of what constitutes a member of the population.

The evolutionary algorithms that are used are rather complicated, incorporating regularization, additional re-ranking, hyper-mutation of hyperparameters etc. It may be that each of these components is necessary and motivated, but it would be really useful to have an ablation study of the components back down to a simple genetic algorithm in at least one setting, to understand how much each component adds (and in which combinations the components must be present).

**Other Comments Or Suggestions:**

I suggest that the authors restructure the paper by moving the Pilot Study to the Appendix, opening up more space for talking about both the positive and negative results of PromptQuine in the main text, for providing ablations on the components of the methods, for improving the description of the evaluation, and for providing more specific qualitative examples of the pruned prompts that work and that don't work well.

Please can the authors comment on the societal implications of their work? In particular, it would be useful to have their thoughts on the implications of this work for interpretability. If LLMs can be prompted for particular behaviors in ways that are not legible by humans, could this be used to hide bad intent? What mitigations might we think about for such problems?

**Other Strengths And Weaknesses:**

N/A

**Questions For Authors:**

Questions are present in my comments above. Please address these and give your thoughts / responses to the specific weaknesses and suggestions I have provided above.

**Relation To Broader Scientific Literature:**

The related work should not be relegated completely to the Appendix, in my opinion. There should at least be a paragraph in the main text giving details of the most relevant prior methods and the ways in which this contribution differs.

Two particular papers that are useful points of comparison are PromptBreeder and Automated Prompt Engineer. Can the authors comment on the reasons why these were not discussed in more detail, and why their results are not used as baselines for PromptQuine? It is hard for me to assess whether their method is stronger / more efficient than APE and PromptBreeder without a numerical comparison. (Note: both papers are cited, but that is not necessarily enough - they seem to be to be directly comparable methods, so it would be useful to compare the performance precisely, in so far as is possible, or to argue why not).

**Theoretical Claims:**

N/A.

---

> ### Author Rebuttal · Authors · 2025-03-29
>
> We sincerely appreciate Reviewer KfU1 for **extremely detailed review** on many tech details. We are glad that the reviewer appreciates the *contributions of our insights towards prompting and interpretability*. We address your questions below:
>
> Anonymous link (AL) for tables/data: https://anonymous.4open.science/r/ughj/KfU1/
> >Evaluation Setup
>
> We agree to restructure paper for clear setups. We'll introduce valid/test separations and what typical objective is when we introduce our algos, and highlight that all follow this. We'll also highlight in section2.2 formalism *Solution Selection*, and make each Exp. clear.
> >PromptQuine results & Pilot Study
>
> * **Results**: We fully agree for transparency. The current format is constrained by page limits. We'll improve. Re RLPrompt vs ours, Quine is imperfect in its representation power. As Appendix E claims, future work can combine both for improvements. We'll highlight in limitations by KRwQ. Additionally, RLPrompt in paper is built upon different templates and token positions (line1115vs1172). *Prefix-only* appending forms a *fairer* comparison: Acc on template1&2: Quine(83.1/79.5), RL(77.5/76.3). The gap bridged, suggesting nuanced sensitivity towards token compositions.
>
> * **Qualitative analysis**: Inspired by the numbers swapping ranks of methods, we'll add a section for ICL stabilization (ICLS). While pruning can be viewed as alternative for ICLS, *template selection [1] can still largely affect results* (Why/when not work effectively). Altogether, along with ablations, this provides a broad view of the pruning formulation.
>
> >Jailbreaking
>
> We clarify we didn't manipulate metrics. Varied setups are limitations in many jailbreaking papers, e.g., own held-out sets for valid/test. We follow [2] by prompting Llama-Guard3 (same prompt,line 405) as classifier on AdvBench [2]. The use of these two is indeed popular [3,4,5+], validating our choices. To increase confidence, we use standard Exact Match score [3] to do the eval: 90+ASR. Please check AL (README) for more numbers, which we reply to e5jB. We also provide **vicuna's direct prediction** and mistral-instruct-v0.3 results in the AL for checks. Inspired by e5jB, we'll also expand jailbreaking, and present a clearer picture of success/failures for that task (see our replies to 8j6A, thanks).
> >Label Word Study
>
> We agree this introduces variances, also mentioned in line 415. We'll expand the texts to make clear. That is also sth always debated by existing work on ICL mechanistics (cf. [6,7] vs [8]). We attempt to present some insights following [8], e.g., intriguing novel finding: pruning has potentials *even with random label words*, which can inspire future work. We'll expand further following ur suggestions on implications.
> >GPQA,MMLU
>
> As MMLU involves many tasks, we show GPQA-main (llama3-it): 33.7 to 35.3 (split done for search&eval). The results are indeed empirical. More research can be done for filling the theoretical gap.
> >ES intro & ablation
>
> We'll follow ur textbook tip. For ablation, many designs are indeed helpful, e.g., stagnation in App. D.7. We'll follow section5 in [9], studying reranking-counter deception/overfit (see AL), mutation rate, offspring size-reward lower-quality stepping stones [10], etc. We'll put significant emphasis on the search dynamics discussion in the paper. The studies can help future research.
> >Promptbreeder(PB)&APE
>
> We excluded PB due to the lack of released code, making faithful reproduction hard. Here, we reimplemented using GPT-4o mutators(1-shot, 3 seeds), each taking 20min for GPT-2 and 30+min for LLaMA3-it(l3it) w/ OpenAI API. Ours outperforms on GPT-2, ~1–7 min and short initializations, but runtime will increase with longer contexts. Given the impact of implementation on search algos [11], we’ll instead claim *decent runtime efficiency* in paper for rigorousness. We chose not to report APE for its constrained exploration vs. PB...
> |PB-Quine|Subj|News|Yahoo|
> |-|-|-|-|
> |l3it|-2.2|-0.6|-1.4|
> |l3it-4shot|-9.5|-1.3|-0.8|
> |GPT2|-5.6|-8.9|-21.5|
>
> PB is strong but limited, e.g., worse in weaker models. UnnaturalLang still good across models-unaddressed.
> >Writing Questions
>
> Thanks for the tips. We'll follow and add implications/limitations/future work. Re interpretability, our findings inspire directions for ICL, e.g., why random labels can improve (start from chance, w/o scaling up shots&sizes [12]) and a rethink on label words in-context. We'll make clear. Re hiding intention, extensive RLHF could address, yet imperfect and we think answering why pruning works can contribute.
>
> [1] start worrying about prompt formatting [2] Understanding jailbreak success [3] GCG [4] AdvPrompter [5] ArtPrompt [6] Ground-Truth Labels Matter [7] ICL Learns Label Relationships yet not.. [8] Rethinking the Role of Demonstrations [9] Large-scale evolution of Image Classifiers [10] Why Greatness cannot be planned [11] Larger LMs do ICL differently [12] Implementation Matters in DPG

---

> > ### Comment · Reviewer_KfU1 · 2025-04-02
> >
> > Many thanks for your rebuttal. On a few points your response lacks detail, so I am not minded to update my score at present.
> >
> > For instance:
> >
> > 1. Can you specifically state the new paragraph that you will insert to explain the train / validation / test split used for Evaluation throughout (and whether it is different for your Pilot Study and for PromptQuine)?
> >
> > 2. Can you be precise about the way you will improve the presentation of the results (i.e. we will present the Figures A,B,C from the Appendix in the main text to show the limitations of PromptQuine)? Can you provide a more detailed paragraph with justification as to why PromptQuine doesn't outperform RLPrompt on these tasks?
> >
> > 3. How exactly are the label words identified?
> >
> > 4. I don't understand the given table comparing PromptBreeder and PromptQuine. Can you provide a table where one column is PromptBreeder, another column is PromptQuine and the rows are different tasks, clearly showing the benefits of PromptQuine?
> >
> > 5. What specific paragraph will you add to discuss the societal implications?
> >
> > Finally, a quick point on style in writing your rebuttal. In my view, it is much more courteous to the reviewers to write your rebuttal using full sentences, correct grammar and avoiding abbreviations.

---

> > > ### Author Response · Authors · 2025-04-03
> > >
> > > We apologize for the word limits and any uncertainty above. We're also preparing a preprint w/ refactorized code.
> > >
> > > Q1: First, we'll write *Solution selection* in section2.3: *Once the search converges or terminates, the algorithm returns a selected optimal solution, i.e., an optimal prompt. The optimality of a prompt is typically assessed using an aggregated metric score on a held-out dataset. This dataset is often referred to as the validation set, while the overall performance is reported separately on an official test set using its specific task metric. We ensure strict separation between the validation and test sets to prevent data leakage and enable a reliable assessment of generalization.* This serves as the foundation for all experiments. Further, rigorously, we won't call those in-search samples (for fitness estimation) as training set, as we don't involve training. For TAPruning/PromptQuine, after considerations, we plan to first illustrate the **Baselines** paragraph and then follow a **Evaluation Settings** paragraph in showing the experimental results. This shall look much better than current formats (mix in one paragraph, e.g., line196/346). Since TAPruning & PromptQuine differ in solution selection (e.g., proxies, rerankings), we'll explain these differences in detail. However, they share key similarities, e.g., the importance of using valid samples with task metrics for final prompt selection. We'll clearly separate the baselines and settings for both in two paragraphs to clarify our setup effectively.
> > >
> > > Q2: We'll first put pilot study in appendix (introduce TAPruning in main paper) and condense texts for GAs. We promise that we'll reorganize section6 (**A Deeper Look into Pruning Effects on ICL**) with another subsection: *On the limitation of PromptQuine*. Specifically, we plan to discuss: *While pruning tokens are effective at enhancing overall ICL results, we identify its inherent limitations for current PromptQuine. Specifically, we observe two recurring failure cases across tasks: (1) token pruning is not a universally reliable method for stabilizing ICL performance, as its effectiveness remains highly sensitive to the chosen ICL templates; (2) fixed-order prompt subsequence search lacks sufficient exploration power for consistently improving performance.* We'll enhance experiments, by including varied ICL templates (e.g., 83 vs 79 in PIQA) and comparing against PB/RLPrompt w/ varied token insertions (e.g. recent tokens) under different tasks, like we did in the first rebuttal. We'll include examples/numbers for justification, e.g., the numbers in first rebuttal. We'll also discuss the failure by selection pressure: some mutations, e.g.,+ tokens, are more effective in long run, yet filtered where novelty search may help.
> > >
> > > Q3: We'll make this clear for section6, where labels in-context are the same as verbalizers. We ensure consistent use of label words (e.g., great/terrible, see Table6) in ICL prompts, following RLPrompt for intuitive verbalizers. Label word identification is based on exact matching, which may introduce slight noise if such words appear in exemplar inputs. However, after further checks, the ICL prompts in the analysis do not contain these words.
> > >
> > > Q4: We organize tables as follows (3 seeds, where PB builds upon original ICL prompts). Unless otherwise stated, PB is built upon 1-shot ICL prompt for Llama3-it, in format <prompt+exemplars>. We're considering adding such numbers in paper.
> > > |Task|PB|PromptQuine|PB (4-shot)|PromptQuine (4-shot)|PB (GPT2)|PromptQuine (GPT2)|
> > > |-|-|-|-|-|-|-|
> > > |Subj|84.3|86.5|83.6|93.1|72.2|77.8|
> > > |AGNews|88.6|89.2|88.3|89.4|57.8|66.7|
> > > |Yahoo|62.8|64.2|65.4|66.2|25.7|47.2|
> > >
> > > Q5: We'll write in Impact statement, beginning with a summary of our work, followed by open interpretability questions, and concluding with a discussion on its societal implications:
> > > *Moreover, we highlight the direct societal implications of our findings on unnatural language. Notably, our work exposes critical weaknesses in current LLM alignment techniques. Despite extensive training designed to align models with human values and ethical standards when given natural language instructions, our findings reveal that unnatural language can still be used to elicit malicious behaviors—exploiting gaps that developers cannot fully anticipate. As demonstrated in our paper, this vulnerability persists even in large models subjected to extensive red teaming. While continuously iterating on red teaming and eliminating failure cases is beneficial, we advocate for exploring novel alignment techniques that go beyond surface-level fixes. In particular, a stronger focus on inner alignment may lead to more robust improvements. For commercial models, we strongly recommend complementing red teaming with output-level restrictions, as this may provide a more intuitive and effective safeguard—especially given that existing alignment methods are primarily optimized for handling natural language inputs.*

---

### Decision · Program_Chairs · 2025-05-01

**Decision:**

Accept (poster)

**Comment:**

In this submission, the authors propose a prompt optimization framework called PromptQuine, an evolutionary search framework that automatically searches for a pruning strategy. This system performs in-context optimization by evolving and refining effective prompts by leveraging only the tokens present within the context. The resulting pruned prompts can often be gibberish but retain some key features, including reference to the task at hand, and demonstrate improvements compared to other baselines.

The reviewers believe that the paper provides an original, significant scientific contribution and shows interesting empirical results. The reviewers appreciated the extensive experiments across various tasks and models which combined provide strong evidence of the method’s effectiveness and generalizability. The reviewers also highlighted several notable weaknesses, such as vague and ambiguous claims and terminology, as well as a lack of clarity throughout the paper. While the authors have addressed these during the rebuttal, I expect these will be incorporated into the final version of the paper. Furthermore, the reviewers also noted limited mechanistic analysis present in the work, which would strengthen the paper.

Overall Assessment: Based on the reviewers' comments and discussions during the rebuttal, I conclude that this paper makes a strong technical contribution and should be accepted. I hope the reviewers’ comments can help the authors prepare an improved version of this submission for camera-ready.